# Does perceived scarcity of COVID-19 vaccines increase vaccination willingness? Results of an experimental study with German respondents in times of a national vaccine shortage

**Julia Schnepf**  *

Department of Social, Economic and Environmental Psychology, University of Koblenz-Landau, Landau, Germany

* schnepf@uni-landau.de

## Abstract

Vaccine shortage is still a major problem in many countries. But how does the vaccine shortage affect people's willingness to be vaccinated? To test whether perceived scarcity of SARS-CoV-2 vaccines has an impact on vaccination willingness, a preregistered online experiment with $N$ = 175 non-vaccinated German participants was conducted during a period of national vaccine shortage. Perceived vaccine scarcity was manipulated by either telling participants that SARS-CoV-2 vaccines in their district would be particularly scarce in the upcoming weeks or that above-average quantities would be available. The results show that individuals in the scarcity-condition were significantly more willing to get vaccinated than those in the surplus-condition. In addition, individuals in the scarcity-condition were found to express more anger towards the debate on relaxations for vaccinated versus non-vaccinated individuals. The results indicate that even superficial processes such as a perception of scarcity can influence people's willingness to get vaccinated.

## Introduction

Scarcity is a well-known mechanism for increasing people's demand for goods [1, 2]. A prominent example of this is the reaction to a 1973 episode of *The Tonight Show*, in which the host, Johnny Carson, jokingly reported a national shortage of toilet paper. Within days of airing, millions of Americans began hoarding toilet paper [3]. Another example is the 1983 Cabbage Patch riots, which were a series of violent riots in stores across America spurred on by the extremely high demand for a newly released line of dolls from Coleco Industries, which were limited in number ([4] cited by [2]).

When people perceive a good to be scarce, they often consider it to be more valuable and are willing to pay more for it [5–8]. More succinctly, the perceived scarcity of a good increases its attractiveness [7, 9, 10]. This has been also demonstrated in a popular study by Worchel and colleagues (1975), in which participants were given either many or a few cookies to try

**Data Availability Statement:** The data and syntax of this study have been made available in the Open Science Framework (https://osf.io/pzury/?view_only=7ebaf5889d3b49cb8e5448c5f2f88b21).

**Funding:** The author(s) received no specific funding for this work.

**Competing interests:** The authors have declared that no competing interests exist.

**Abbreviations:** COVID-19, Coronavirus Disease 2019; IBM SPSS, International Business Machines Statistical Package for the Social Sciences; SARS-CoV-2, Severe Acute Respiratory Syndrome Coronavirus 2.

[11]. When participants were offered only a few, this increased their desire for more cookies. The effect became even stronger when participants were told there were only a few cookies left because others had eaten them. Similar effects have been found in various shopping situations, showing that customers tend to give greater preference to products of which only a limited number is available ([12], see also [13], [14] for limited supply effects in conspicuous consumption). Gupta and Gentry (2019) showed that perceived scarcity even led to an increase in a customer's intention to hide a particular product in a store, so that they could come back and buy it later [15]. In addition, participants reported higher anticipated regret from not buying a product which they perceived to be scarce.

As for the context of vaccines, the term "scarcity mentality" was first used by Gregory Poland, chief of the vaccine research group at the Mayo Clinic in Rochester, MN, US, during the 2004 nationwide shortage of flu vaccines [16]. He observed that this shortage massively increased the vaccination willingness of healthcare workers. The scarcity situation affected "(. . .) even those who [had] been adamantly opposed to taking the vaccine (. . .)" (cited by [16], p. 715). Data from a large-scale field experiment among hospital employees by De Janvry and colleagues (2006) empirically confirmed the impression of a scarcity effect on flu vaccination in 2004 [17]. Information provided to hospital staff about the shortage of flu vaccines increased respondents' demand by 110%.

Currently, shortage effects have been also evident with regard to the supply of SARS-CoV-2 vaccines. At first, those countries that ordered large quantities of SARS-CoV-2 vaccines at the early stages of the vaccine development went on to have some of the world's highest vaccination rates (e.g., Israel, the US, and the UK) [18]. Policy makers in the European Union, on the other hand, were more hesitant with their vaccine purchasing decisions [19], which went on to create a scarcity situation in the first half of 2021 [20].

In addition to facing a low supply of vaccines, Germany especially had a high administrative burden, and a dysfunctional infrastructure, as well as a slow start of its national COVID-19 vaccination campaign [21]. At the same time, Germany started with only moderate vaccination acceptance rates compared to other European countries and a general hesitancy towards the newly developed SARS-CoV-2 vaccines [22]. Moreover, the debate over possible side effects of the AstraZeneca vaccine shook the vaccination willingness of many people [23].

However, despite all these initial obstacles, the overall level of vaccination willingness increased–particularly during the period of the national vaccine shortage in Germany in the first half of the 2021 [24], [25]. To test whether the perceived scarcity of SARS-CoV-2 vaccines has an increasing effect on the vaccination willingness of German respondents, a preregistered online experiment was conducted in which information on the local availability of SARS-CoV-2 vaccines was varied. I hypothesize that participants' vaccination willingness increases if they are presented with a fictitious statement about an upcoming local shortage versus surplus of SARS-CoV-2 vaccine (Main Hypothesis).

In addition to the preregistered Main Hypothesis, I was also interested in whether information on SARS-CoV-2 vaccine scarcity increases the anger of non-vaccinated individuals towards the debate over relaxations for vaccinated (versus non-vaccinated) citizens. In Germany, this debate first arose in February 2021, when a sizable portion of the prioritized groups (mainly high age groups) had already been vaccinated. However, the debate became particularly heated and polarized between April and May 2021, when relaxations for vaccinated people were legally adopted despite the fact that a majority of the population had still not yet been vaccinated due to the nationwide shortage. In national opinion polls, many expressed their opposition to these relaxations [26]. I therefore aim to exploratively test whether scarcity (versus surplus) information increases people's anger towards this debate (Exploratory Research Question).

## Methods

### Participants and procedure

Research question, design, minimum sample size, exclusion criteria, and analysis plan of the study were preregistered on AsPredicted previous to the data collection (see https://aspredicted.org/x2yg4.pdf). Participants were recruited between May 07 and 19, 2021 via a German crowdsourcing platform [27] and were compensated with EUR 0.60 for a maximum duration of 3 minutes. Invitation mails were sent out by the crowdsourcing platform and have been automatically weighted by the age and gender distribution of the German population. At the beginning of the online survey, participants confirmed the informed consent. As the treatment contained mild deception, the informed consent contained information that they might be deceived during participation and that the true purpose of the study could not be explained until the end. Then participants were asked whether they already have a fixed vaccination appointment or have been vaccinated at least once. For participants who answered "yes" or "don't know" to one of these questions, the survey ended, so that only non-vaccinated individuals without an existing vaccination appointment participated in the experiment. This was followed by questions on sociodemographic characteristics. Participants were also asked to indicate the county (German: Landkreis) in which they currently live, as the COVID-19 vaccination campaign in Germany was primarily managed by county. This was a filler question to make the treatment more credible. Then, participants were randomly assigned to a short text claiming either that their county would be provided with a below average amount of vaccine in the coming weeks or an above average amount (scarcity vs. surplus condition). The county-level manipulation was chosen because people are more likely to be well informed about the vaccine shortage at the national level, but less so about local supplies, as those are less often reported in the media. The text read as follows:

> According to a press release from the Department of Health, your county is expected to receive significantly fewer [more] doses of the COVID-19 vaccine than other counties will in the coming weeks. Consequently, a particularly high shortage [availability] of vaccines is expected in your region.

Participants then answered questions on the dependent variables and the manipulation check, namely how good or bad they perceived the supply situation of their county. At the end of the survey, respondents were thanked for their participation and were informed of the intention of the study. Since deception took place, participants were not directed to the last survey page including the participation code for the payment until they indicated that they had read and understood the debriefing information. This has been implemented for ethical reasons, so that participants would be informed about the true purpose of the study in any case.

### Measures

**Dependent variables.** Participants' vaccination willingness was measured by the mean value of the following three items: "Will you get vaccinated as soon as possible?"; "I will be trying harder to get a vaccination appointment in the near future."; "If necessary, I will use various channels (e.g., primary physician and vaccination centers) to get a vaccination appointment."; reaching from 1 = "very unlikely" to 7 = "very likely", Cronbach's $\alpha$ = .93, $M$ = 4.65, $SD$ = 1.98. The naming of the dependent variables differs from the preregistration. In the preregistration, vaccination willingness and appointment intention were mentioned separately. For reasons of conceptual overlap and high internal consistency, the items were combined into one general vaccination willingness scale, as the use of single item measures is increasingly

criticized if multi-item measures are possible [28]. Also, item three was not explicitly mentioned in the preregistration as the preregistration only included a reference to example items of each scale.

For the explorative analysis, participants were also asked to answer whether they are angered by the debate over relaxations for vaccinated individuals ("I am angry about the current debate over advantages and liberties for vaccinated people," 1 = "completely disagree" to 7 = "completely agree", $M$ = 4.44, $SD$ = 2.12). This item was added for explorative reasons, but was not part of the preregistration.

**Manipulation check.** Perceived vaccine availability was measured with a single item: "How do you feel about the supply situation for COVID-19 vaccines in your county?" ranging from 1 = "very poor" to 7 = "very good", $M_{overall}$ = 3.66, $SD$ = 1.27.

## Sample size and data analysis

Since scarcity effects were found to be of large rather than small effect sizes in previous studies (see [9, 11, 14]), a small sample size was assumed to be sufficient to identify scarcity effects on vaccination willingness. This is why a minimum sample size of 120 participants was mentioned in the preregistration. In addition, an a-priori power analysis using G*Power (version 3.1, [28]) for the detection of a medium ($d$ = .50 to .70) to large ($d$ = .80 to $\geq$ 1, [29]) scarcity effect in a Multivariate Analysis of Variance (MANOVA) including two dependent variables with $d$ = .60, 1-β = .95, α = .05 resulted in a required sample size of 158 participants. To allow for potential dropout, data from 238 individuals was collected. A one-way MANOVA with treatment (scarcity vs. surplus) as an independent factor, and vaccination willingness and anger towards the debate over relaxations for vaccinated citizens as dependent variables was preregistered.

## Exclusion criteria

Following the recommendations of Leiner (2019), participants with a high relative speed index (RSI > 2), which is a measurement provided by the survey platform used and is based on processing time that identifies suspicious data patterns associated with poor quality, and reading times of equal to or less than 2 $SD$ below the sample mean were excluded from data analysis [30, 31]. There is a growing body of criticism on doing experimental research without controlling for a successful experimental manipulation within the sample under research [32, 33]. Given that people's prior knowledge and information could have strongly influenced whether people believed the treatment texts or not, I preregistered the manipulation check variable as an exclusion criterion, as done in other experimental research [34–36]. Thus, participants for whom the scarcity vs. surplus manipulation did not work, i.e., individuals in the scarcity condition who perceived the availability to be high (perceived vaccine availability > 4) and individuals in the surplus condition who perceived the availability to be low (perceived vaccine availability < 4), were excluded from the data analysis. The exact sample sizes of the excluded sub-groups as well as their characteristics are reported in a CONSORT flow diagram in the online supplement (S1 File). The final sample size consisted of 175 participants (108 male, 67 female; $M_{age}$ = 35.93, $SD$ = 9.88; education level: 43.4% university degree, 28% baccalaureate, 20.6% completed apprenticeship; 7.4% other secondary school certificate). The sample was not representative for the entire German population. But, this was not the intention of the study. More importantly, the sample was intended to correspond to the characteristics of the non-vaccinated population at the time of the survey. According to the German Federal Statistical Office, the average age of the sample under research roughly corresponds to that of the unvaccinated population in May 2021 [36]. There is no available official data for the distribution of

gender and educational attainment among non-vaccinated individuals in Germany, at the field time of the survey (see [37]). However, later data on vaccination intentions has shown that women tend to be rather unvaccinated than men [38], indicating a potential over-representation of male respondents in the sample under research.

## Results

The IBM Statistical Package for the Social Sciences (SPSS) 26 was used for the statistical analyses. Prior to the main analysis, I tested whether the experimental conditions varied significantly regarding the distribution of sociodemographic characteristics to control for possible biases. Overall, $n = 98$ respondents were assigned to the scarcity condition and $n = 77$ to the surplus condition. The sub-samples did not significantly differ regarding participants' age, $M_{\text{age scarcity}} = 36.02$, $SD = 9.80$, $M_{\text{age surplus}} = 35.81$, $SD = 10.05$, $t(173) = 0.14$, $p = .887$, gender, $\chi^2 (1, N = 175) = 0.23$, $p = .642$, nor educational attainment, $\chi^2 (6, N = 175) = 3.18$, $p = .786$. With these results, systematic biases between the treatment conditions can be excluded, which allows potential treatment effects to be attributed to the experimental manipulation. In addition, results of a $t$-test between the scarcity and surplus condition regarding the manipulation check variable revealed that individuals in the scarcity condition perceived the supply situation as significantly worse than those in the surplus condition, $M_{\text{scarcity}} = 2.93$, $SD = 1.10$, $M_{\text{surplus}} = 4.58$, $SD = 0.78$, $t(172) = -11.14$, $p < .001$.

To compare both experimental treatment conditions, a one-way MANOVA was conducted. Results showed a small but significant scarcity effect on both dependent variables (with both effect sizes $.010 < \eta^2 < .039$ indicating a small effect [29]). In line with the Main Hypothesis, participants in the scarcity condition reported a significantly higher willingness to get vaccinated against the SARS-CoV-2 virus compared to those in the surplus condition, $M_{\text{scarcity}} = 4.91$, $SD = 1.88$, $M_{\text{surplus}} = 4.31$, $SD = 2.07$, $F(1,174) = 4.03$, $p = .043$, $\eta^2 = .023$ (see Fig 1). Also with regard to the Exploratory Research Question, I found a significant difference between both conditions, indicating higher anger towards the debate over lockdown relaxations for vaccinated versus non-vaccinated individuals among participants who were presented with the scarcity (versus surplus) information, $M_{\text{scarcity}} = 4.75$, $SD = 1.98$, $M_{\text{surplus}} = 4.05$, $SD = 2.24$, $F(1,174) = 4.77$, $p = .030$, $\eta^2 = .027$ (see Fig 1). The overall model test revealed that the treatment differences were also significant on the combination of dependent variables, $F(2, 171) = 5.91$ $p = .003$, $\eta_p^2 = .065$, Wilk's $\lambda = .935$, $1-\beta = .87$, indicating a significant explanation of variance on both dependent variables by the experimental treatment. The same statistical model was also calculated with age, gender, and education level as covariates, to test the robustness of

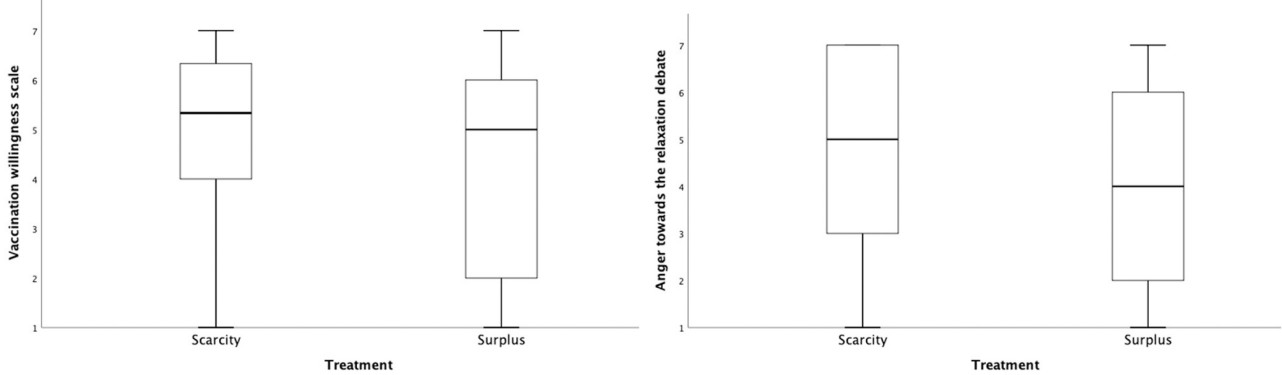

**Fig 1. Vaccination willingness and anger towards the relaxation debate as a function of treatment.**

the results when controlling for additional control variables. The results remained the same in both models. The model including covariates is presented in Table 1 of the supplements (S2 File). In addition, the results remained stable when conducting the same analysis on the level of the single items instead of the scale (S3 File) and when the sample was weighted for participants gender, using a weighting variable which represents the actual gender distribution among vaccination sceptics at the field time of the survey (S4 File).

## Discussion

The study was embedded in the SARS-CoV-2 vaccination situation in Germany. The results support the hypothesis that scarcity information on the supply of SARS-CoV-2 vaccines in Germany has an increasing effect on the vaccination willingness of non-vaccinated individuals. This is an interesting finding given that, in the course of the discussions about the efficacy and possible side effects of the AstraZeneca vaccine in particular, many experts feared that the general vaccination willingness in Germany would drop dramatically (see expert assessment by [39], p. 62). However, the willingness to get vaccinated against the SARS-CoV-2 virus, regardless of which vaccine was administered, has sharply increased in the German population when vaccine shortage was high. In fact, when some cities announced they would be freely distributing their remaining AstraZeneca vaccines in 2021, thousands of people went to the vaccination centers a full day in advance and waited in kilometer-long lines just to get one of the coveted doses [40]. This has left the impression that the scarcity of SARS-CoV-2 vaccines may increase the willingness of individuals to be vaccinated. However, this observation at the macro level did not allow for any causal conclusions on the micro level of vaccination behavior [41]. For this reason, an experimental study was conducted to test whether a scarcity impression of the SARS-CoV-2 vaccine can indeed lead to an increased willingness to vaccinate.

The results of the presented study support the idea that perceived scarcity of SARS-CoV-2 vaccines may be one driving factor for increasing people's vaccination willingness. Given the simplicity of the design and the rather small size of the effects found, of course, only limited conclusions about the complexity of vaccination decisions can be drawn from such a small study, and a cross-sectional experiment like this one only represents a momentary snapshot, through which it is not possible to construct a comprehensive explanation of the processes of the last weeks and months. Against this background, it also has to be mentioned here that a significant number of participants was excluded as the experimental manipulation has not worked for them, i.e., their perceived availability of COVID-19 vaccines was not affected by the experimental condition. On one hand, using manipulation checks as an exclusion criterion guarantees that the participants under research have been influenced by the given information in the intended direction, which is a necessary precondition to draw any causal conclusions between the experimental variation and the variation on the dependent variables [32, 33]. On the other hand, this practice is increasingly criticized in some social science disciplines [42] because of the risk of increasing biases between the sub-samples. As in this study, there were no significant differences between the tested sub-samples regarding their sociodemographic composition, the risk of potential bias in this study is comparatively small.

Nevertheless, it has to be critically mentioned here that the high number of excluded cases due to a failed manipulation check may speak for the fact that some participants might not have completely read or understood the information, or that they were highly informed about the issue so that the presented information probably has not affected their perception of vaccine availability. This is a clear limitation of the presented study and can only be solved in the future by (a) choosing more distinctive texts to induce scarcity or surplus impressions, (b) conducting quasi-experimental field experiments in which local scarcity situations are covered as

independent variables, as well as (c) cross-sectional research on the effects of temporal vaccine scarcity on individuals' vaccination willingness.

That said, the results of this study do fit into a larger picture of scarcity effects in varying fields and confirm that scarcity information has a promoting effect on the demand of a good [12–14, 17]. Another question is whether such effects are desired in the vaccination context. On the one hand, in the fight against the COVID-19 pandemic, it is highly desirable to achieve herd immunity as quickly as possible by maximizing vaccination rates. On the other hand, scarcity effects represent a rather superficial form of information processing (see [7, 9]). In this vein, Pereira and colleagues (2021) found that perceived scarcity of SARS-CoV-2 vaccines increased vaccination willingness among people with low compassion for others, but decreased it among those with high compassion for others [43]. Scarcity induced vaccination willingness may disappear the moment vaccines become widely available. From an informed patient perspective, it would therefore be more desirable to increase vaccination willingness through deeper persuasion [44, 45].

The results of this study also revealed that participants in the scarcity compared to the surplus condition expressed greater anger towards the debate over liberties and relaxations for vaccinated versus non-vaccinated individuals. Such anger may also quickly turn into reactance and a reduced willingness to follow COVID-19 rules [46]. This is also in line with findings of a later study by Sprengholz and colleagues (2022) showing that reactance towards COVID-19 measures was higher among people with a high willingness to vaccinate when they were told that vaccines would be scarce in the future [47]. The task of policymakers must therefore be to deal responsibly with vaccine shortages and to communicate priorities and measures in such a way that they are met with the broadest possible acceptance.

## Conclusion

SARS-CoV-2 vaccines are now highly available–at least in the global north [18]. Nevertheless, temporary and local shortages are still likely to occur in the future, especially with regard to booster vaccinations and vaccines that are adapted to new variants of the SARS-CoV-2 virus. Against this background, it is particularly important that governments communicate such shortages in a responsible manner, as this is the only way to ensure that the prioritization of vulnerable groups during shortages remains accepted by the general public [43].

## Ethics statement

Data collection was conducted in accordance with the ethical standards of the German Association of Psychology (DGPS) and with the 1964 Helsinki Declaration. The study was approved by the local ethics committee of the University of Koblenz-Landau (agreement number LEK-374). All participants agreed with the privacy statement and the informed consent of the study.

## Supporting information

**S1 File. CONSORT diagram and sub-sample comparisons.**
(DOCX)

**S2 File. Robustness check of the results: Controlling for covariates.**
(DOCX)

**S3 File. Robustness check of the results: Single item analysis.**
(DOCX)

**S4 File. Robustness check of the results: Using gender weight.**
(DOCX)

## Author Contributions

**Conceptualization:** Julia Schnepf.

**Data curation:** Julia Schnepf.

**Formal analysis:** Julia Schnepf.

**Investigation:** Julia Schnepf.

**Methodology:** Julia Schnepf.

**Writing – original draft:** Julia Schnepf.

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
