## [Decision Letter · Decision Letter 0]

22 Dec 2021

PONE-D-21-30612Does perceived scarcity of COVID-19 vaccines increase vaccination willingness? Results of an experimental study with German respondents in times of a national vaccine shortage.PLOS ONE

Dear Dr. Schnepf,

Thank you for submitting your manuscript to PLOS ONE. After careful consideration, we feel that it has merit but does not fully meet PLOS ONE’s publication criteria as it currently stands. Therefore, we invite you to submit a revised version of the manuscript that addresses the points raised during the review process. Please revise the paper in accordance to the reviewers' comments. In the revised version of the paper, please better state the aspects related to sampling and validation of the model. 

We look forward to receiving your revised manuscript.

Kind regards,

Camelia Delcea

Academic Editor

PLOS ONE

Journal Requirements:

Reviewers' comments:

Reviewer's Responses to Questions

**Comments to the Author**

1. Is the manuscript technically sound, and do the data support the conclusions?

Reviewer #1: Partly

Reviewer #2: Yes

Reviewer #3: Yes

Reviewer #4: Partly

Reviewer #5: Partly

Reviewer #6: Partly

2. Has the statistical analysis been performed appropriately and rigorously? 

Reviewer #1: No

Reviewer #2: Yes

Reviewer #3: I Don't Know

Reviewer #4: No

Reviewer #5: Yes

Reviewer #6: No

3. Have the authors made all data underlying the findings in their manuscript fully available?

Reviewer #1: No

Reviewer #2: Yes

Reviewer #3: Yes

Reviewer #4: No

Reviewer #5: Yes

Reviewer #6: No

4. Is the manuscript presented in an intelligible fashion and written in standard English?

Reviewer #1: Yes

Reviewer #2: Yes

Reviewer #3: Yes

Reviewer #4: Yes

Reviewer #5: Yes

Reviewer #6: No

5. Review Comments to the Author

Reviewer #1: This is a well written and concise paper reporting a useful survey of the impact of Covid vaccine scarcity on the likelihood of unvaccinated individuals to get vaccinated.

The author should detail the number of participants that were excluded according to each of the exclusion criteria. This could either be in the text or in a CONSORT-type diagram.

Page 4: ‘especially Germany’ would read better as ‘Germany especially’

Top of page 6: ‘have a fix vaccination appointment’ - ‘fix’ should be replaced with ‘fixed’

Page 6, near the bottom’: Since deception took place, participants were not directed to the participation code until they indicated that they had read and understood the debriefing information.’ - it’s not clear what the ‘participation code’ is.

Top of page 7: the author should define the alpha that is given here.

Sample size calculation: please define the parameters.

Results: please define the parameters (M, SD, etc).

Has the author assessed the validity of the assumptions underlying MANOVA, eg. Normality of the response? It seems likely to me that they will not hold and that a nonparametric alternative may be necessary.

The author should provide some interpretation and discussion of the magnitude of the differences between the groups. The difference in means was less than 1 between the groups for both outcomes and I would argue that this actually a very small difference - perhaps even to the extent that this is a positive thing and may mean that people are relatively likely (both groups >4) to get vaccinated regardless of the availability of the vaccine. I think there is a nuance between the statistical and societal significance of the results that has been missed by the author.

Figure 1 should be displayed as box plots instead of the bar plots shown. This would help with the interpretation of the results, as referred to in the previous point, since it will give an overview of the entirety of each group.

Reviewer #2: -the author should be removed "see e.g." from the citations (see e.g. , Westdeutscher Rundfunk, May 2021). ).

-the author should be included the list of abbreviations eg. WHAT does MANOVA, IBM, SPSS and COVID-19 means?

-write the formula and assumption of MANOVA?

- the methods of this study like lecture notes so rewrite again

Reviewer #3: Dear author,

The abstract missing the conclusion part, there are still characters left to add that part.

Was validity of the survey assessed?

Many of the references used here are quiet old, additionally, it is unreasonable to compare the scarcity of condiments to that of medical procedures.

Reviewer #4: This short paper experimentally tests whether information about COVID-19 vaccine scarcity increases vaccine willingness in Germany in May 2021. Participants were recruited online for a short survey their manipulated (with deception) whether respondents were informed that their county was or was not experiencing vaccine shortages, before assessing the impact of this treatment on immediate self-reported willingness to vaccinate as well as anger about restrictions on the unvaccinated. The results show about a 0.3 standard deviation increase in willingness to vaccinate, suggesting – in line with studies relating to non-COVID vaccines – that scarcity can increase willingness to get vaccinated.

While the paper is concise and speaks broadly to a key topic – determinants of vaccine uptake – there are also substantial limitations in terms of what can be learned from this study. As noted in detail below, the writeup is missing important information, there are reasons to doubt the value of the findings, and the theoretical mechanisms are not explored.

The concision of the paper is good, but it goes too far. It is actually quite difficult to evaluate the study due to a lack of standard information about the study. In particular:

• None of the assumptions required for valid inference are supported empirically. There are no tests validating that treatment is balanced across predetermined covariates (location and demographic variables, in this case), which is important in studies like this with a small sample. Moreover, there are no tests of differential attrition – do people exposed to different messages answer the main outcome questions at different rates? If so, is the sample that survives to the outcome questions still well balanced?

• The author notes that survey invitations were sent in order to match the age and gender population distribution. However, this does not tell us what the sample that actually takes the survey looks like – is it representative on these dimensions or any other dimensions that were not matched on? Ultimately, we know very little about the sample, making it hard to evaluate whether this online sample is representative of the German population (or other populations that we might want to extrapolate to).

• A manipulation check was conducted, which is good, but the results of it are not reported. Did the manipulation work, and for how many people did it work? This is important because it’s hard to gauge whether the intensity of treatment is – i.e. how many people changed their minds because of treatment. This also has implications for whether the mechanism driving the results is belief updating or increasing salience.

• The main vaccination intention outcome is comprised of 3 survey questions. However, the author does not report how the 3 items are combined – is the main outcome the average of the 3, an inverse-covariance weighted scale, a factor, or something else? In addition, it would be nice to see the results reported by item separately, since the second and especially the first outcomes seem to be the most relevant ones for understanding vaccination intentions, but may not be the outcomes that are driving treatment effects.

• It is also worth noting that the preregistration information deviates somewhat from the final paper. The sample size differs, another outcome is introduced in the paper, the scale for vaccination outcomes is not mention in the analysis plan, the definition of treatment failing was not defined in advance, etc. These are not major issues, but some justification for deviation should be provided.

Despite the experimental nature of the study, I have several concerns about the design itself:

• First, the outcomes are vaccine intentions measured immediately after a manipulation that respondents were told was being done to them. This raises important concerns about social desirability biases. Did people thinking vaccines were scarce feel the need to answer that they will get one, out of shame or experimenter demand? To be fair, most studies in the literature have used survey intentions – rather than behavioral outcomes – to measure vaccination, but this is increasingly unsatisfactory given that vaccines are now widely available (and were fairly available in Germany in May 2021 too) and because this particular survey started by telling participants “that they might be deceived during participation and that the true purpose of the study could not be explained until the end.”

• Second, the exclusion criteria seem to risk post-treatment bias. While excluding “speeders” is fine, I worry about excluding people on the basis of their posterior beliefs about scarcity. It is unorthodox in my field to condition the sample (or use covariate adjustment) on basis of post-treatment covariate, as this introduces biases, see e.g. Hernan and Robins 2011. I would encourage the author not to use this potentially bias-inducing exclusion restriction. If they wish to normalize by the degree to which the manipulation works, they would be better served using an instrumental variables regression. This should be easy to implement in a revision, although it’s not clear how robust the results would be.

• Third, it wasn’t clear to me whether the treatment would be interpreted by respondents as reflecting low supply of or high demand for vaccines. The former sentence in the treatment suggests a supply mechanism, but the latter could be either. This has implications for interpreting the results.

Finally, I had several concerns regarding the broader contribution:

• The experiment has very little to say about what theoretical mechanisms might be driving the scarcity effect. I think a variety of mechanisms, with differing policy implications, could be at play – is scarcity altering perceptions of the value of vaccination, due to limited number or social learning (i.e. high uptake suggests others think vaccines are good)? Is scarcity creating pressures to conform because people are learning that others around them are getting vaccinated? Is scarcity simply galvanizing people into action, but without altering their valuations or social incentives to vaccinate?

• Especially if we can’t say much about the mechanisms, the policy implications are not clear. Moreover, it seems impractical and unsustainable to suggest that governments should maintain low vaccine supplies at all times or that they should lie to people about supplies being low. So, while it’s useful to know that low availability affect vaccine uptake, its seem like an inherently transient factor with few policy implications or aggregate uptake implications.

Reviewer #5: The article describes a short online experiment to determine if perceived scarcity of COVID-19 vaccines has an impact on vaccination willingness in Germany, during May 2021. The experiment was pre-registered, and provides access to the final data set, the pre-registered exclusion rules, and the SPSS syntax to redo the calculations, which makes a strong case for the rigourosity of the analysis.

I have minor comments about the manuscript that I'm going to list as follows:

1) You are reporting Wilks Lambda and Eta square, but you are not interpreting nor commenting on those results in your discussion. Perhaps it would be better to add a short interpretation in both cases.

2) Add a comparison between your final sample (N = 175) demographics with German demographics, and add in your discussion a comment on possible biases that the online experiment might have. One of them that is clear to me, it's the participation of more men than women in this study. Are there any references about possible biasses induced by gender?. Also, comment on the limitations and advantages of using a paid website to recruit participants. Are there any references about possible biases on this self recruited population? If so, I think it's important to add them to the manuscript.

3) For Table 1, report exact p-values along with the Eta square for treatment. This is the main result of the analysis, hence saying p<0.05 or p<0.01 may not be enough.

4) A descriptive plot for the Vaccination willingness and Anger measures according to each of the covariates could be useful to describe the data set before entering into the analysis.

Reviewer #6: Summary

The study explores the association between the perceived scarcity of the COVID-19 vaccine and people’s willingness to get vaccinated. The topic is very timely and is of significant importance. However, there are a number of issues that need to clarified about the study design to ensure its validity and about its limitations and the effect of these limitations on the results and their interpretation. To account for potential confounders and quantify the effect size accurately, I strongly recommend Author perform multivariate regression analysis, instead of one-way MANOVA. If this cannot be done, Author should explain the reasons in the discussion as it seems to be a better study design to examine the association between perceived scarcity of vaccines and willingness to get vaccinated. To be acceptable for publication, the manuscript requires substantial revision as follows.

Introduction

Author described in detail the impact of vaccine scarcity on consumer demand and its applicability to the current context of COVID-19 vaccine roll-out. In general, the introduction described the current situation in Germany well to provide context to the study. However, all these factors mentioned in the introduction should be accounted for in the analysis. Also, there is little justification for the reason why Author decided to measure the “anger” as an exploratory analysis (i.e., why is it of importance and what is its relevant to the main hypothesis?).

Page 3, paragraph starting with “As for the context of vaccines, …”: This sentence does not appear to have the correct citation. Pls check and include an appropriate citation.

Page 4, sentence starting with “At first, those countries…” requires citation.

Page 4, sentence starting with “However, despite all these initial obstacles, …”: When you say “the period of the national vaccine shortage”, please be explicit and state the exact period in the text.

Page 5, paragraph starting with “In addition to the test of a scarcity…”: This is the first time Author introduced the debate over relaxation of measures for those vaccinated. If this is one of the outcome measures you aim to test in your study, please provide more context as to its importance before introducing a question related to it.

Methods

Fundamentally, I am concerned with the validity of the study design.

1) Author should not describe this study as an experimental study. There is no counterfactual (pre-/post-test or treatment vs. control) group or randomization. The study appears to be a cross-sectional study.

2) The ‘treatment’ described in the study is not a true treatment. The instant exposure to crafted information on vaccine surplus/scarcity is unlikely to lead to an immediate change in participants’ perception about vaccination. Therefore, exposure to treatment -> associated outcome relationship here is not established.

3) Similarly, the study design does not rule out the possibility that the participants’ willingness to be vaccinated was established prior to their exposure to information on vaccine surplus/scarcity. There is no pre-/post- comparison to establish the temporal relationship between the exposure and the outcome. The current results without this does not support a causal relationship.

4) In addition, Author used one-way MANOVA to test the hypothesis, which does not account for the potential confounding effects. This way, it cannot be ruled out that the participants’ willingness to be vaccinated is driven/mediated by other factors than the perceived scarcity of vaccines. This contradicts what was stated in the introduction and the discussion that vaccine efficacy or side effects also can drive people’s willingness to be vaccinated. Author needs to justify why one-way MANOVA was used on the set of outcome variables. Author should also separately examine and present in a table whether there are any groups differences with regard to demographic and other confounding variables to bolster the findings.

5) Lastly, it is unclear why Author did not choose to perform multivariate regression analysis on each of the outcome variables to account for potential confounding factors. This should be explained in the discussion.

Currently, your methods section is a lengthy walk-through of the online survey. Some information you included, such as how participants were “thanked for their participation”, is not essential. Rather than describing the survey procedure in detail, please provide the well-articulated summary of what you were trying to measure, how they were measured in the survey (i.e. type of question or scale used), and how they were coded in the data. Also, whenever a technical term is introduced, spell it out the first time it was used, and explain what it is used for.

Page 5, sentence starting with “Invitation mails were sent out…”: What was the sampling pool for this invitation? what is meant by “weighted by” age and gender?

Page 6, paragraph starting with “Participants then answered questions…”: What are the dependent variables and what do you mean by ‘manipulation check’? What do you mean when participants were not directed to the “participation code”? What is the “debriefing information”? Are these important facts to be mentioned in this section?

Page 7, “Dependent variables”:

1) What was your rationale to include these three questions to measure the outcomes? Are these questions from a validated instrument to measure willingness to receive vaccination? (If yes, please cite the work.)

2) Why is the scale from 1 – 7? It appears to be unconventional.

3) Did you combine the three questions to form one outcome measure scale? If so, how did you combine them? (i.e., average of 3, sum of 3, etc.).

4) What is the ‘alpha’? (if it is Cronbach’s alpha, you need to specify it explicitly in the text and explain what it means)

Page 7, “Manipulation check”:

What is the purpose of measuring this variable?

Page 7, “Sample size and data analysis”

1) In the first sentence, please clarify what a “large effect size” is and a “small effect size” is. Please provide appropriate ranges.

2) For the power calculation, please specify d, 1-beta, alpha. Also, please spell-out MANOVA when it appears for the first time in the text.

3) In the last sentence, what do you mean when you say “A one-way MANOVA …. was pre-registered?”

Page 7, “Exclusion criteria”

1) What is a relative speed index? Please describe.

2) Spell out SD when it appears for the first time in the text.

3) The description of the final sample for the study should be provided in the “Results” section using a table including all the variables used in the analysis. Please provide such a table in this section.

Results

1) Currently the results section lacks a descriptive table of the sample (a conventional table).

2) In addition, the way in which the results is reported in the text is hard to follow. A table for the results of the one-way MANOVA analysis for each group should be included. Author listed a list of abbreviated terms with figures without explaining what they mean. There is no effect size reported for the treatment, or no justification why the different tests were used to report the results.

Discussion

Overall, your discussion is based on the findings from your study to draw general inference on the German population. I am concerned on the validity of the discussion in general because 1) the generalizability of the findings based on a small sample of 175 participants is not discussed; and 2) the current study design does not allow any causal inference. Limitations should also expand further to accommodate the concerns raised in the “Methods” section.

Page 8, sentence starting with “The results support the hypothesis…”: Based on the current study design, there is very little evidence generated from the study to support this statement.

Page 9, sentence starting with “Surprisingly, this has not..”: You cannot say this without testing for this. Your current study does not explore the association between the mentioned factors and your dependent variable.

Page 9, sentence starting with “In fact, when some cities…”: This is irrelevant to the findings of this study and should rather presented in the introduction.

Page 9, sentence starting with “On the other hand, scarcity mentalities..”: This sentence is vague and does not add any value to the discussion as it is formed. Author should clarify it.

Page 9, sentence starting with ”Such anger may also quickly…”: Is there any literature documenting such an association? If yes, Author should cite it. This could be an interesting mechanism and Author should clearly state this in the introduction to justify the exploratory question (focusing on measurement of anger).

Minor revisions:

No lines numbers provided in the submission file. This made the review of this manuscript difficult which will also make its revision similarly difficult.

Referencing style does not comply with PLOS ONE’s submission guideline, and in-text citation format is inconsistent throughout (i.e., avoid adding hyperlink to a webpage directly in the text, use a citation software so that in-text citations are numbered in Vancouver style).

Footnotes are not permitted as per PLOS One’s guidelines.

Overall, writing needs to be improved in terms of grammar and style. For example, inconsistent use of tense throughout the text; inconsistent use of terminology on “COVID-19 vaccine” (vs. SARS-CoV-2 vaccine)

6. PLOS authors have the option to publish the peer review history of their article (what does this mean?). If published, this will include your full peer review and any attached files.

Reviewer #1: **Yes: **Dr SJE Barry

Reviewer #2: **Yes: **Yenew Alemu

Reviewer #3: No

Reviewer #4: No

Reviewer #5: No

Reviewer #6: No

---

## [Author Response · Author response to Decision Letter 0]

24 Feb 2022

Dear Dr. Delcea,

I would like to resubmit the revised version of the manuscript entitled “Does perceived scarcity of COVID-19 vaccines increase vaccination willingness? Results of an experimental study with German respondents in times of a national vaccine shortage.” (PONE-D-21-30612)

I would like to thank you and the reviewers for your assessment and evaluation of the manuscript! 

I made some major revisions which I shortly want to outline here. First, I have added additional information in the supplements regarding the sample characteristics of the excluded participants by adding a CONSORT diagram as asked for by reviewer 1. Second, I have added additional analyses on the single items as raised by reviewer 4 mainly showing the same pattern of results. Third, I have added an additional test of whether the sample is balanced regarding sociodemographic information which is the case. Also, I added an additional t-test on whether the manipulation check has worked across the whole sample – which also was the case. Last, according to concerns raised by reviewer 4 and 6, I have lowered the tone in the discussion a bit so that the results of the study do not seem to be overestimated.

I have tried to implement all addressable points in the revision and used clearer language in many parts of the manuscript to overcome possible misunderstandings. With regard to some questions on the experimental method or, for example, the question of what a manipulation check is, which has been raised in some of the reviews, I have attempted to clear them by giving brief answers in my response letter.

Overall, I want to highlight that this is a short report on the results of a small preregistered study on scarcity effects in the context of the COVID-19 pandemic. As such one, to my experience and opinion, it has several strengths which I am not used to typically see in this category of research articles. First, the study has been preregistered to the data collection and and the results support the preregistered hypothesis. Second, the preregistration, data, and syntax have been immediately made public after data collection in order to meet the open science criteria. Third, the short report deals with a (still) highly topical issue, as a next wave of vaccination shortage can be expected when the omicron-specific SARS-CoV-2 vaccines will be available.

Attached is a revised version of the manuscript with tracked changes and an untracked version.

In this response letter, I have bolded my responses to the reviewers’ comments.

Thank you for giving me the chance to having revised and resubmitted the manuscript! 

Best,

The author

5. Review Comments to the Author

Reviewer #1: This is a well written and concise paper reporting a useful survey of the impact of Covid vaccine scarcity on the likelihood of unvaccinated individuals to get vaccinated.

The author should detail the number of participants that were excluded according to each of the exclusion criteria. This could either be in the text or in a CONSORT-type diagram.

I thank reviewer 1 for their overall evaluation of the manuscript and I agree that a CONSORT diagram is useful given the exclusion of respondents who failed the manipulation check. This is why the sample description was revised and a CONSORT Flow Chart was added in the online supplements (l. 160-174):

“The exact sample sizes of the excluded sub-groups as well as their characteristics are reported in a CONSORT flow diagram in the online supplement (S1). The final sample size consisted of 175 participants (108 male, 67 female; Mage = 35.93, SD = 9.88; education level: 43.4% university degree, 28% baccalaureate, 20.6% completed apprenticeship; 7.4% other secondary school certificate). The sample is not representative for the entire German population. But, this was not the intention of the study. More importantly, the sample was intended to correspond to the characteristics of the unvaccinated population at the time of the survey. According to the German Federal Statistical Office, the average age of the sample under research roughly corresponds to that of the unvaccinated population in May 2021 (Statistisches Bundesamt, 2021). There is no available data for the distribution of gender and educational attainment among non-vaccinated individuals in Germany, at the field time of the survey (see Impfdashboard of the Robert-Koch-Institute, 2022). However, later data on vaccination intentions has shown that women tend to be rather unvaccinated than men (e.g., Graeber et al., 2021), indicating a potential over-representation of male respondents in the sample under research.”

Page 4: ‘especially Germany’ would read better as ‘Germany especially’

Thanks for this language advice! It has been changed in the revised manuscript.

Top of page 6: ‘have a fix vaccination appointment’ - ‘fix’ should be replaced with ‘fixed’

Again, thanks a lot, I changed it.

Page 6, near the bottom’: Since deception took place, participants were not directed to the participation code until they indicated that they had read and understood the debriefing information.’ - it’s not clear what the ‘participation code’ is.

I agree that this first read a bit unspecific. I revised it in the new version of the manuscript (l. 119-123):

“Since deception took place, participants were not directed to the last survey page including the participation code for the payment until they indicated that they had read and understood the debriefing information. This has been implemented for ethical reasons, so that the participants would be informed about the true purpose of the study in any case.”

Top of page 7: the author should define the alpha that is given here.

Sample size calculation: please define the parameters.

Results: please define the parameters (M, SD, etc).

I agree with reviewer 1 and added the additional information (l. 127-143):

“Dependent variables. Participants’ vaccination willingness was measured the mean value of the following three items: “Will you get vaccinated as soon as possible?”; “I will be trying harder to get a vaccination appointment in the near future.”; “If necessary, I will use various channels (e.g., primary physician and vaccination centers) to get a vaccination appointment.”; reaching from 1 = “very unlikely” to 7 = “very likely”, Cronbach’s α = .93, M = 4.65, SD = 1.98. The naming of the dependent variables differs from the preregistration. In the preregistration, vaccination willingness and appointment intention were mentioned separately. For reasons of conceptual overlap and high internal consistency, the items were combined into one scale.

For the explorative analysis, participants were also asked to answer whether they are angered by the debate over relaxations for vaccinated individuals (“I am angry about the current debate over advantages and liberties for vaccinated people,” 1 = “completely disagree” to 7 = “completely agree”, M = 4.44, SD = 2.12).

Manipulation check. Perceived vaccine availability was measured with a single item: “How do you feel about the supply situation for COVID-19 vaccines in your county?” ranging from 1 = “very poor” to 7 = “very good”, Moverall = 3.66, SD = 1.27.”

The information about the sample size estimation was changed as follows (l. 147-151):

“An a-priori power analysis using G*Power (version 3.1, Faul et al., 2009) for the detection of a medium to large scarcity effect in a Multivariate Analysis of Variance (MANOVA) including two dependent variables with d = .60, 1-β = .95, α = .05 resulted in a required sample size of 158 participants. To allow for potential dropout, data from 238 individuals was collected.”

Has the author assessed the validity of the assumptions underlying MANOVA, eg. Normality of the response? It seems likely to me that they will not hold and that a nonparametric alternative may be necessary.

Yes, the assumptions for conducting a MANOVA have been assessed. Homogeneity of variances within both treatment conditions was given, whereas the distribution on the response variables differed somewhat from normal distribution. I want to argue that there are several reasons why it is plausible that the dependent variables deviate from normal distribution. First, since the beginning of the pandemic, vaccination has been a very polarized issue resulting in many people being either pro or contra COVID-19 vaccines, which is also reflected by the data. Second, at the time of the survey, there was more uncertainty about the existing vaccines, their safety, and possible side effects, which may have also influenced the distribution of the dependent variables. However, the (M)ANOVA is known to be a statistical tool which tends to be especially robust towards violations of the normal distribution assumption (Blanca et al., 2017; Schmider et al., 2010).

Nevertheless, the analysis was repeated with box-cox transformed dependent variables yielding the same pattern of results as reported in the paper – with and without the inclusion of covariates (https://osf.io/pzury/?view_only=7ebaf5889d3b49cb8e5448c5f2f88b21). 

The author should provide some interpretation and discussion of the magnitude of the differences between the groups. The difference in means was less than 1 between the groups for both outcomes and I would argue that this actually a very small difference - perhaps even to the extent that this is a positive thing and may mean that people are relatively likely (both groups >4) to get vaccinated regardless of the availability of the vaccine. I think there is a nuance between the statistical and societal significance of the results that has been missed by the author.

The effect size was mentioned referring to the range of small effect sizes (l. 217-219):

“Results showed a small but significant scarcity effect on both dependent variables (with both effect sizes .010 < η2 < .039 indicating a small effect, see Lenhard & Lenhard, 2016).”

Also the discussion was revised in order to relativize the findings in accordance with the effect sizes observed (l. 284-290):

“(…) This has left the impression that the scarcity of SARS-CoV-2 vaccines may increase the willingness of individuals to be vaccinated. However, this observation at the macro level did not allow for any causal conclusions on the micro level of vaccination behavior (Ramström, 2018). For this reason, an experimental study was conducted to test whether a scarcity impression of the SARS-CoV-2 vaccine can indeed lead to an increased willingness to vaccinate.

 The results of the presented study support the idea that perceived scarcity of SARS-CoV-2 vaccines may be one driving factor for increasing people’s vaccination willingness. Of course, only limited conclusions about the complexity of vaccination decisions can be drawn from such a small study, and a cross-sectional experiment like this one only represents a momentary snapshot, through which it is not possible to construct a comprehensive explanation of the processes of the last weeks and months. 

That said, the results of this study do fit into a larger picture of scarcity effects in varying fields and confirm that scarcity information has a promoting effect on the demand of a good (Aggarwal et al., 2011; De Janvry et al., 2006; Gierl & Huettl, 2010; Jang et al., 2015). Another question is whether such effects are desired in the vaccination context. On the one hand, in the fight against the COVID-19 pandemic, it is highly desirable to achieve herd immunity as quickly as possible by maximizing vaccination rates. On the other hand, scarcity effects represent a rather superficial form of information processing (see, e.g., Mittone & Savadori, 2009; Verhallen, 1982). In this vein, Pereira and colleagues (2021) found that perceived scarcity of SARS-CoV-2 vaccines increased vaccination willingness among people with low compassion for others, but decreased it among those with high compassion for others. Scarcity induced vaccination willingness may disappear the moment vaccines become widely available. From an informed patient perspective, it would therefore be more desirable to increase vaccination willingness through deeper persuasion (see, e.g., Argote et al., 2021; Gowda et al., 2013).”

Figure 1 should be displayed as box plots instead of the bar plots shown. This would help with the interpretation of the results, as referred to in the previous point, since it will give an overview of the entirety of each group.

Again, thanks for this advice! Figure 1 has been replaced by a boxplot.

Reviewer #2: -the author should be removed "see e.g." from the citations (see e.g. , Westdeutscher Rundfunk, May 2021). 

-the author should be included the list of abbreviations eg. WHAT does MANOVA, IBM, SPSS and COVID-19 means?

I think for most researchers in the field these are frequently used abbreviations which are typically used in scientific papers. According to APA guidelines, it is absolutely standard to abbreviate the data analysis software, such as SPSS or R.

Nevertheless, I changed it in the revised version of the manuscript (l. 147-150, l. 179-180):

“An a-priori power analysis using G*Power (version 3.1, Faul et al., 2009) for the detection of a medium to large scarcity effect in a Multivariate Analysis of Variance (MANOVA) including two dependent variables with d = .60, 1-β = .95, α = .05 resulted in a required sample size of 158 participants.”

“The IBM Statistical Package for the Social Sciences (SPSS) 26 was used for the statistical analyses.”

-write the formula and assumption of MANOVA?

A MANOVA is a standard model (not only) in psychological research providing a general test of whether the variance of certain response variables differs to a higher extent between experimental conditions than within experimental conditions. It is very unusual (not recommended by the reporting guidelines of the field) to precise this by any formula. I know, this may be more the case in fields of economic modelling or when using regression models in other disciplines, but it is typically not done when using a standard model of comparing two experimental conditions with each other.

See Nyhus Hagum et al. (2022), Roznik & Alford (2015), or Tricarico et al. (2011) for some examples of PLOS one articles using a MANOVA without specifying the underlying mathematical formular (which would be also odd for the underlying study).

- the methods of this study like lecture notes so rewrite again

Also in line with some comments raised by reviewer 4, I revised many parts of the methods section of the paper (l. 119-177).

Reviewer #3: Dear author,

The abstract missing the conclusion part, there are still characters left to add that part.

Thank you for this advice! I have followed reviewer 3’s recommendation and added a conclusion part to the abstract.

Was validity of the survey assessed?

The validity of the survey is mainly expressed by the internal consistency of the scales used in the questionnaire. These were on an extremely satisfying level (Cronbach’s alpha > .80, see Cortina, 1993). 

Many of the references used here are quiet old, additionally, it is unreasonable to compare the scarcity of condiments to that of medical procedures. 

Again, thanks for the recommendation! I have added more recent references (also in the context of COVID-19 vaccines). 

e.g.:

Hajek, K. V., & Häfner, M. (2021). Paradoxes of Reactance During the COVID-19 Pandemic: A Social-psychological Perspective. Javnost-The Public, 28(3), 290-305. https://doi.org/10.1080/13183222.2021.1969619

Pereira, B., Fehl, A. G., Finkelstein, S. R., Jiga‐Boy, G. M., & Caserotti, M. (2021). Scarcity in COVID‐19 vaccine supplies reduces perceived vaccination priority and increases vaccine hesitancy. Psychology & Marketing. https://doi.org/10.1002/mar.21629

Sprengholz, P., Betsch, C., & Böhm, R. (2021). Reactance revisited: Consequences of mandatory and scarce vaccination in the case of COVID‐19. Applied Psychology: Health and Well‐Being, 13(4), 986-995. https://doi.org/10.1111/aphw.12285

However, generally spoken, I would nevertheless argue that, first, research is not like cheese that is going bad with time (an analogy which I was taught by my supervisor), second, the psychological processes that lead people to find scarce goods more attractive may be the same across different classes of products and are likely to be the same regardless of the time in which people live.

Reviewer #4: This short paper experimentally tests whether information about COVID-19 vaccine scarcity increases vaccine willingness in Germany in May 2021. Participants were recruited online for a short survey their manipulated (with deception) whether respondents were informed that their county was or was not experiencing vaccine shortages, before assessing the impact of this treatment on immediate self-reported willingness to vaccinate as well as anger about restrictions on the unvaccinated. The results show about a 0.3 standard deviation increase in willingness to vaccinate, suggesting – in line with studies relating to non-COVID vaccines – that scarcity can increase willingness to get vaccinated.

While the paper is concise and speaks broadly to a key topic – determinants of vaccine uptake – there are also substantial limitations in terms of what can be learned from this study. As noted in detail below, the writeup is missing important information, there are reasons to doubt the value of the findings, and the theoretical mechanisms are not explored.

The concision of the paper is good, but it goes too far. It is actually quite difficult to evaluate the study due to a lack of standard information about the study. In particular:

• None of the assumptions required for valid inference are supported empirically. There are no tests validating that treatment is balanced across predetermined covariates (location and demographic variables, in this case), which is important in studies like this with a small sample. Moreover, there are no tests of differential attrition – do people exposed to different messages answer the main outcome questions at different rates? If so, is the sample that survives to the outcome questions still well balanced?

I thank reviewer 4 for their advice! It is true, that (although conducted) there was no report of a systematic test between the both treatment groups. This was added with regard to the relevant covariates (l. 174-183):

“Prior to the main analysis, I tested whether the experimental conditions varied significantly regarding the distribution of sociodemographic characteristics to control for possible biases. Overall, n = 98 respondents were assigned to the scarcity condition and n = 77 to the surplus condition. The sub-samples did not significantly differ regarding participants’ age, Mage scarcity = 36.02, SD = 9.80, Mage surplus = 35.81, SD = 10.05, t(173) = 0.14, p = .887, nor gender, χ2 (1, N = 175) = 0.23, p = .642, nor educational attainment, χ2 (6, N = 175) = 3.18, p = .786. With this result, systematic biases between treatment conditions can be excluded, which allows potential treatment effects to be attributed to the experimental manipulation.”

I did not analyze the dropout rates of the entire participation, because the Survey Platform defaults to downloading only complete data sets and to directing only participants who have completed the questionnaire to the final participation code. However, in this study I don't see any theoretical benefit of analyzing the dropout rates either. I find it more important that the final treatment groups are approximately equal in size and balanced with respect to sociodemographic variables. This is the case as described above.

• The author notes that survey invitations were sent in order to match the age and gender population distribution. However, this does not tell us what the sample that actually takes the survey looks like – is it representative on these dimensions or any other dimensions that were not matched on? Ultimately, we know very little about the sample, making it hard to evaluate whether this online sample is representative of the German population (or other populations that we might want to extrapolate to).

Again, I agree with reviewer 4’s concerns and added further information on the representativity of the sample under research. In this context, it is important to note that the aim of the study was not to investigate scarcity effects among a sample which was representative for the general population, but among a sample which was roughly representative for the population of non-vaccinated individuals at the time of the study. Therefore, the following section was added (l. 162-172):

“The sample is not representative for the entire German population. But, this was not the intention of the study. More importantly, the sample was intended to correspond to the characteristics of the unvaccinated population at the time of the survey. According to the German Federal Statistical Office, the average age of the sample under research roughly corresponds to that of the unvaccinated population in May 2021 (Statistisches Bundesamt, 2021). There is no available data for the distribution of gender and educational attainment among non-vaccinated individuals in Germany, at the field time of the survey (see Impfdashboard of the Robert-Koch-Institute, 2022). However, later data on vaccination intentions has shown that women tend to be rather unvaccinated than men (Graeber et al., 2021), indicating a potential over-representation of male respondents in the sample under research.”

• A manipulation check was conducted, which is good, but the results of it are not reported. Did the manipulation work, and for how many people did it work? This is important because it’s hard to gauge whether the intensity of treatment is – i.e. how many people changed their minds because of treatment. This also has implications for whether the mechanism driving the results is belief updating or increasing salience.

I agree with reviewer 4’s comment. I added information on the t-test comparing both treatment groups on the MC variable (l. 192-195):

“In addition, results of a t-test between the scarcity and surplus condition regarding the manipulation check variable revealed that individuals in the scarcity condition perceived the supply situation as significantly worse than those in the surplus condition, Mscarcity = 2.93, SD = 1.10, Msurplus = 4.58, SD = 0.78, t(172) = -11.14, p < .001.”

Please note that, although not reported in the paper, this difference on the MC variable kept robust when no exclusion criteria have been used:

Mscarcity = 3.37, SD = 1.41, Msurplus = 3.81, SD = 1.32, t(234) = -2.43, p = .016

• The main vaccination intention outcome is comprised of 3 survey questions. However, the author does not report how the 3 items are combined – is the main outcome the average of the 3, an inverse-covariance weighted scale, a factor, or something else? In addition, it would be nice to see the results reported by item separately, since the second and especially the first outcomes seem to be the most relevant ones for understanding vaccination intentions, but may not be the outcomes that are driving treatment effects.

Thanks for this advice! I also added a separate MANOVA on the single items in the supplements which mainly show the same pattern of results (S3).

• It is also worth noting that the preregistration information deviates somewhat from the final paper. The sample size differs, another outcome is introduced in the paper, the scale for vaccination outcomes is not mention in the analysis plan, the definition of treatment failing was not defined in advance, etc. These are not major issues, but some justification for deviation should be provided.

Yes, I agree with reviewer 4’s comment on the deviation from the preregistration. This was now made more clear in the revised version of the manuscript, e.g.:

“The naming of the dependent variables differs from the preregistration. In the preregistration, vaccination willingness and appointment intention were mentioned separately. For reasons of conceptual overlap and high internal consistency, the items were combined into one general vaccination willingness scale.”

“This item was added for explorative reasons, but was not part of the preregistration.”

“This is why a minimum sample size of 120 participants was mentioned in the preregistration. In addition, an a-priori power analysis using G*Power (version 3.1, Faul et al., 2009) for the detection of a medium (d = .50 to .70) to large (d = .80 to \\geq 1, Lenhardt & Lenhardt, 2016) scarcity effect in a Multivariate Analysis of Variance (MANOVA) including two dependent variables with d = .60, 1-β = .95, α = .05 resulted in a required sample size of 158 participants. To allow for potential dropout, data from 238 individuals was collected.”

But, excluding participants for whom the treatment has not worked has been previously preregistered as an exclusion criterion: “participants for whom the treatment failed“. (https://aspredicted.org/blind.php?x=kt8us2) 

Despite the experimental nature of the study, I have several concerns about the design itself:

• First, the outcomes are vaccine intentions measured immediately after a manipulation that respondents were told was being done to them. This raises important concerns about social desirability biases. Did people thinking vaccines were scarce feel the need to answer that they will get one, out of shame or experimenter demand? To be fair, most studies in the literature have used survey intentions – rather than behavioral outcomes – to measure vaccination, but this is increasingly unsatisfactory given that vaccines are now widely available (and were fairly available in Germany in May 2021 too) and because this particular survey started by telling participants “that they might be deceived during participation and that the true purpose of the study could not be explained until the end.”

I understand reviewer 4’s concerns. But, I have to disagree a little bit regarding the statement that COVID-19 vaccines have been fairly available in May 2021 in Germany. The effective vaccination rate has been around 40 % at the time of the experiment (see https://impfdashboard.de/en/) and COVID-19 vaccines have been scarce (see e.g., Warren & Lofstedt, 2021 for a discussion). In addition, many citizens were disappointed by the vaccination campaign of the German government regarding the high bureaucratic burden and ongoing supply problems. 

The advantage of the treatment, however, at this time, was that people could not really assess whether their county would be one of those with many vaccine doses or few vaccine doses in the future. Therefore, the treatment tied into this point and the scarcity impression was manipulated by saying that the participant’s county would be either among the poorly supplied (scarcity) or among the well supplied (surplus).

With regard to informed consent, according to the local ethics commission of my university, it must always include information that participants may be deceived if deception is involved in the study. However, the informed consent form is very long and the question arises whether it is fully recalled by participants or whether the information about deception is perceived as one of many other pieces of the informed consent, and thus may not have been recalled by them. In addition, the manipulation check variable guaranteed that only participants were included in the final data analysis who believed in the presented information. This is the strongest guarantee that can be provided in order to avoid pure desirability effects (Fiedler et al., 2021). Also, the manipulation check was measured after the assessment of the dependent variables which also excludes the risk that this measure biases the results in a certain direction (Kidd, 1976).

• Second, the exclusion criteria seem to risk post-treatment bias. While excluding “speeders” is fine, I worry about excluding people on the basis of their posterior beliefs about scarcity. It is unorthodox in my field to condition the sample (or use covariate adjustment) on basis of post-treatment covariate, as this introduces biases, see e.g. Hernan and Robins 2011. I would encourage the author not to use this potentially bias-inducing exclusion restriction. If they wish to normalize by the degree to which the manipulation works, they would be better served using an instrumental variables regression. This should be easy to implement in a revision, although it’s not clear how robust the results would be.

At least in social psychological research it is common practice to define exclusion criteria for extremely short and long processing times. Both are related to low data quality (see Leiner, 2019 for a discussion). Relatedly, Draisma and Dijkstra (2004) found that longer latencies are related to higher response error rates. However, there were no additional participants being excluded by the speed variables – they were already part of the subsample excluded by the failed manipulation. So there is no additional bias by this exclusion criteria (nevertheless, I mentioned them because they were part of the preregistration).

• Third, it wasn’t clear to me whether the treatment would be interpreted by respondents as reflecting low supply of or high demand for vaccines. The former sentence in the treatment suggests a supply mechanism, but the latter could be either. This has implications for interpreting the results.

The scarcity impression was mainly manipulated by telling participants that their county will face a below-average supply. The experimental manipulation read as follows:

“According to a press release from the Department of Health, your county is expected to receive significantly fewer [more] doses of the COVID-19 vaccine than other counties will in the coming weeks. Consequently, a particularly high shortage [availability] of vaccines is expected in your region.“

Finally, I had several concerns regarding the broader contribution:

• The experiment has very little to say about what theoretical mechanisms might be driving the scarcity effect. I think a variety of mechanisms, with differing policy implications, could be at play – is scarcity altering perceptions of the value of vaccination, due to limited number or social learning (i.e. high uptake suggests others think vaccines are good)? Is scarcity creating pressures to conform because people are learning that others around them are getting vaccinated? Is scarcity simply galvanizing people into action, but without altering their valuations or social incentives to vaccinate?

• Especially if we can’t say much about the mechanisms, the policy implications are not clear. Moreover, it seems impractical and unsustainable to suggest that governments should maintain low vaccine supplies at all times or that they should lie to people about supplies being low. So, while it’s useful to know that low availability affect vaccine uptake, its seem like an inherently transient factor with few policy implications or aggregate uptake implications.

I have aimed to address both concerns by re-writing the discussion section of the manuscript:

“(…) This has left the impression that the scarcity of SARS-CoV-2 vaccines may increase the willingness of individuals to be vaccinated. However, this observation at the macro level did not allow for any causal conclusions on the micro level of vaccination behavior (Ramström, 2018). For this reason, an experimental study was conducted to test whether a scarcity impression of the SARS-CoV-2 vaccine can indeed lead to an increased willingness to vaccinate.

 The results of the presented study support the idea that perceived scarcity of SARS-CoV-2 vaccines may be one driving factor for increasing people’s vaccination willingness. Of course, only limited conclusions about the complexity of vaccination decisions can be drawn from such a small study, and a cross-sectional experiment like this one only represents a momentary snapshot, through which it is not possible to construct a comprehensive explanation of the processes of the last weeks and months. 

That said, the results of this study do fit into a larger picture of scarcity effects in varying fields and confirm that scarcity information has a promoting effect on the demand of a good (Aggarwal et al., 2011; De Janvry et al., 2006; Gierl & Huettl, 2010; Jang et al., 2015). Another question is whether such effects are desired in the vaccination context. On the one hand, in the fight against the COVID-19 pandemic, it is highly desirable to achieve herd immunity as quickly as possible by maximizing vaccination rates. On the other hand, scarcity effects represent a rather superficial form of information processing (see, e.g., Mittone & Savadori, 2009; Verhallen, 1982). In this vein, Pereira and colleagues (2021) found that perceived scarcity of SARS-CoV-2 vaccines increased vaccination willingness among people with low compassion for others, but decreased it among those with high compassion for others. Scarcity induced vaccination willingness may disappear the moment vaccines become widely available. From an informed patient perspective, it would therefore be more desirable to increase vaccination willingness through deeper persuasion (see, e.g., Argote et al., 2021; Gowda et al., 2013).”

Reviewer #5: The article describes a short online experiment to determine if perceived scarcity of COVID-19 vaccines has an impact on vaccination willingness in Germany, during May 2021. The experiment was pre-registered, and provides access to the final data set, the pre-registered exclusion rules, and the SPSS syntax to redo the calculations, which makes a strong case for the rigourosity of the analysis.

I have minor comments about the manuscript that I'm going to list as follows:

1) You are reporting Wilks Lambda and Eta square, but you are not interpreting nor commenting on those results in your discussion. Perhaps it would be better to add a short interpretation in both cases.

Thanks for the advice! Both points were addressed by adding further information:

„Results showed a small but significant scarcity effect on both dependent variables (with both effect sizes .010 < η2 < .039 indicating a small effect, see Lenhard & Lenhard, 2016).“

“The overall model test revealed that the treatment differences were also significant on the combination of dependent variables, F(2, 171) = 5.91 p = .003, ηp2 = .065, Wilk’s \\mathbit{\\lambda} = .935, 1-β = .87, indicating a significant explanation of variance on both dependent variables by the experimental treatment.”

2) Add a comparison between your final sample (N = 175) demographics with German demographics, and add in your discussion a comment on possible biases that the online experiment might have. One of them that is clear to me, it's the participation of more men than women in this study. Are there any references about possible biasses induced by gender?. Also, comment on the limitations and advantages of using a paid website to recruit participants. Are there any references about possible biases on this self recruited population? If so, I think it's important to add them to the manuscript.

I also added further information on the sample characteristics (l. 198-213):

“The exact sample sizes of the excluded sub-groups as well as their characteristics are reported in a CONSORT flow diagram in the online supplement (S1). The final sample size consisted of 175 participants (108 male, 67 female; Mage = 35.93, SD = 9.88; education level: 43.4% university degree, 28% baccalaureate, 20.6% completed apprenticeship; 7.4% other secondary school certificate). The sample was not representative for the entire German population. But, this was not the intention of the study. More importantly, the sample was intended to correspond to the characteristics of the non-vaccinated population at the time of the survey. According to the German Federal Statistical Office, the average age of the sample under research roughly corresponds to that of the unvaccinated population in May 2021 (Statistisches Bundesamt, 2021). There is no available data for the distribution of gender and educational attainment among non-vaccinated individuals in Germany, at the field time of the survey (see Impfdashboard of the Robert-Koch-Institute, 2022). However, later data on vaccination intentions has shown that women tend to be rather unvaccinated than men (Graeber et al., 2021), indicating a potential over-representation of male respondents in the sample under research.“

“Prior to the main analysis, I tested whether the experimental conditions varied significantly regarding the distribution of sociodemographic characteristics to control for possible biases. Overall, n = 98 respondents were assigned to the scarcity condition and n = 77 to the surplus condition. The sub-samples did not significantly differ regarding participants’ age, Mage scarcity = 36.02, SD = 9.80, Mage surplus = 35.81, SD = 10.05, t(173) = 0.14, p = .887, nor gender, χ2 (1, N = 175) = 0.23, p = .642, nor educational attainment, χ2 (6, N = 175) = 3.18, p = .786. With these results, systematic biases between the treatment conditions can be excluded, which allows potential treatment effects to be attributed to the experimental manipulation. In addition, results of a t-test between the scarcity and surplus condition regarding the manipulation check variable revealed that individuals in the scarcity condition perceived the supply situation as significantly worse than those in the surplus condition, Mscarcity = 2.93, SD = 1.10, Msurplus = 4.58, SD = 0.78, t(172) = -11.14, p < .001.“

3) For Table 1, report exact p-values along with the Eta square for treatment. This is the main result of the analysis, hence saying p<0.05 or p<0.01 may not be enough.

The exact p-values have been added now (l. 205-220):

“The sub-samples did not significantly differ regarding participants’ age, Mage scarcity = 36.02, SD = 9.80, Mage surplus = 35.81, SD = 10.05, t(173) = 0.14, p = .887, nor gender, χ2 (1, N = 175) = 0.23, p = .642, nor educational attainment, χ2 (6, N = 175) = 3.18, p = .786. With these results, systematic biases between the treatment conditions can be excluded, which allows potential treatment effects to be attributed to the experimental manipulation. In addition, results of a t-test between the scarcity and surplus condition regarding the manipulation check variable revealed that individuals in the scarcity condition perceived the supply situation as significantly worse than those in the surplus condition, Mscarcity = 2.93, SD = 1.10, Msurplus = 4.58, SD = 0.78, t(172) = -11.14, p < .001.

To compare both experimental treatment conditions, a one-way MANOVA was conducted. Results showed a small but significant scarcity effect on both dependent variables (with both effect sizes .010 < η2 < .039 indicating a small effect, see Lenhard & Lenhard, 2016). In line with the Main Hypothesis, participants in the scarcity condition reported significantly higher willingness to get vaccinated against the SARS-CoV-2 virus compared to those in the surplus condition, Mscarcity = 4.91, SD = 1.88, Msurplus = 4.31, SD = 2.07, F(1,174) = 4.03, p = .043, η2 = .023 (see figure 1).“

4) A descriptive plot for the Vaccination willingness and Anger measures according to each of the covariates could be useful to describe the data set before entering into the analysis.

This has been also added (l. 139-152):

“Dependent variables. Participants’ vaccination willingness was measured by the mean value of the following three items: “Will you get vaccinated as soon as possible?”; “I will be trying harder to get a vaccination appointment in the near future.”; “If necessary, I will use various channels (e.g., primary physician and vaccination centers) to get a vaccination appointment.”; reaching from 1 = “very unlikely” to 7 = “very likely”, Cronbach’s α = .93, M = 4.65, SD = 1.98. The naming of the dependent variables differs from the preregistration. In the preregistration, vaccination willingness and appointment intention were mentioned separately. For reasons of conceptual overlap and high internal consistency, the items were combined into one general vaccination willingness scale.

For the explorative analysis, participants were also asked to answer whether they are angered by the debate over relaxations for vaccinated individuals (“I am angry about the current debate over advantages and liberties for vaccinated people,” 1 = “completely disagree” to 7 = “completely agree”, M = 4.44, SD = 2.12). This item was added for explorative reasons, but was not part of the preregistration.”

Reviewer #6: Summary

The study explores the association between the perceived scarcity of the COVID-19 vaccine and people’s willingness to get vaccinated. The topic is very timely and is of significant importance. However, there are a number of issues that need to clarified about the study design to ensure its validity and about its limitations and the effect of these limitations on the results and their interpretation. To account for potential confounders and quantify the effect size accurately, I strongly recommend Author perform multivariate regression analysis, instead of one-way MANOVA. If this cannot be done, Author should explain the reasons in the discussion as it seems to be a better study design to examine the association between perceived scarcity of vaccines and willingness to get vaccinated. To be acceptable for publication, the manuscript requires substantial revision as follows.

Introduction

Author described in detail the impact of vaccine scarcity on consumer demand and its applicability to the current context of COVID-19 vaccine roll-out. In general, the introduction described the current situation in Germany well to provide context to the study. However, all these factors mentioned in the introduction should be accounted for in the analysis. Also, there is little justification for the reason why Author decided to measure the “anger” as an exploratory analysis (i.e., why is it of importance and what is its relevant to the main hypothesis?).

Page 3, paragraph starting with “As for the context of vaccines, …”: This sentence does not appear to have the correct citation. Pls check and include an appropriate citation.

The citation was added in l. 48.

Page 4, sentence starting with “At first, those countries…” requires citation.

Ritchie et al., 2020 was added as an appropriate citation in this context (l. 58).

Page 4, sentence starting with “However, despite all these initial obstacles, …”: When you say “the period of the national vaccine shortage”, please be explicit and state the exact period in the text.

There is no concrete time range when the shortage began or ended, but based on further publications on this phenomenon, I added the information that this took place in the first half of the year (l. 70-72).

„However, despite all these initial obstacles, the overall level of vaccination willingness increased – particularly during the period of the national vaccine shortage in Germany in the first half of the 2021 (ARD-Deutschlandtrend, 2021; Cassel & Ulrich, 2021).“

Page 5, paragraph starting with “In addition to the test of a scarcity…”: This is the first time Author introduced the debate over relaxation of measures for those vaccinated. If this is one of the outcome measures you aim to test in your study, please provide more context as to its importance before introducing a question related to it.

As outlined in the manuscript – which is a short report – this was an explorative question and has now been embedded in the following information on why this variable was added (l. 87-92):

“In Germany, this debate first arose in February 2021, when a sizable portion of the prioritized groups (mainly high age groups) had already been vaccinated. However, the debate became particularly heated and polarized between April and May 2021, when relaxations for vaccinated people were legally adopted despite the fact that a majority of the population had still not yet been vaccinated due to the nationwide shortage. In national opinion polls, many expressed their opposition to these relaxations (see Tagesschau, 2021).”

Methods

Fundamentally, I am concerned with the validity of the study design.

1) Author should not describe this study as an experimental study. There is no counterfactual (pre-/post-test or treatment vs. control) group or randomization. The study appears to be a cross-sectional study.

I see that it may have been not clear enough in the procedure section that the study clearly was an experiment as participants were randomly assigned to one of the experimental conditions. This was clarified by the following part (l. 106-111): 

“Then, participants were randomly assigned to a short text claiming either that their county would be provided with a below average amount of vaccine in the coming weeks or an above average amount (scarcity vs. surplus condition). The county-level manipulation was chosen because people are more likely to be well informed about the vaccine shortage at the national level, but less so about local supplies, as those are less often reported in the media.”

2) The ‘treatment’ described in the study is not a true treatment. The instant exposure to crafted information on vaccine surplus/scarcity is unlikely to lead to an immediate change in participants’ perception about vaccination. Therefore, exposure to treatment -> associated outcome relationship here is not established.

Typically, in psychological research experimental conditions are also referred to as “treatments” (see Myers & Hansen, 2012, p. 22). To guarantee that the experimental manipulation of participants’ level of perceived scarcity has worked, a manipulation check was performed after the assessment of the dependent variables. This allows to attribute detected differences on the dependent variables to the previous experimental treatment (Fiedler et al., 2021) – especially when the sample is balanced on further variables, such as socio-demographic control variables, which is the case in the underlying study. This was added by the following section (l. 174-183):

“Prior to the main analysis, I tested whether the experimental conditions varied significantly regarding the distribution of sociodemographic characteristics to control for possible biases. Overall, n = 98 respondents were assigned to the scarcity condition and n = 77 to the surplus condition. The sub-samples did not significantly differ regarding participants’ age, Mage scarcity = 36.02, SD = 9.80, Mage surplus = 35.81, SD = 10.05, t(173) = 0.14, p = .887, nor gender, χ2 (1, N = 175) = 0.23, p = .642, nor educational attainment, χ2 (6, N = 175) = 3.18, p = .786. With this result, systematic biases between treatment conditions can be excluded, which allows potential treatment effects to be attributed to the experimental manipulation.”

3) Similarly, the study design does not rule out the possibility that the participants’ willingness to be vaccinated was established prior to their exposure to information on vaccine surplus/scarcity. There is no pre-/post- comparison to establish the temporal relationship between the exposure and the outcome. The current results without this does not support a causal relationship.

The experimental method, in particular, is the most superior method for enabling causal inference. Unlike correlational data, for example, experiments allow changes in the dependent variable to be attributed to the experimental manipulation. In particular, if it was controlled for that the experimental treatment was successful and if the samples are balanced on additional variables, as is both the case in the present study. I would like to quote Myers and Hansen (2012, p. 22-23) at this point to illustrate what the experimental method means in psychology:

“A psychology experiment is a controlled procedure in which at least two different treatment conditions are applied to subjects. The subjects’ behaviors are then measured and compared to test a hypothesis about the effects of those treatments on behavior. Note that we must have at least two different treatments so that we can compare behavior under varied conditions and observe the way behavior changes as the treatment conditions change.” 

And p. 24:

“The greatest value of the psychology experiment is that, within the experiment, we can infer a cause-and-effect relationship between the antecedent conditions and the subjects’ behaviors. If the XYZ set of antecedents always leads to a particular behavior, whereas other treatments do not, we can infer that XYZ causes the behavior.”

As all these criteria are met by the presented study, this is why the study can be clearly categorized as an experiment.

4) In addition, Author used one-way MANOVA to test the hypothesis, which does not account for the potential confounding effects. This way, it cannot be ruled out that the participants’ willingness to be vaccinated is driven/mediated by other factors than the perceived scarcity of vaccines. This contradicts what was stated in the introduction and the discussion that vaccine efficacy or side effects also can drive people’s willingness to be vaccinated. Author needs to justify why one-way MANOVA was used on the set of outcome variables. Author should also separately examine and present in a table whether there are any groups differences with regard to demographic and other confounding variables to bolster the findings.

5) Lastly, it is unclear why Author did not choose to perform multivariate regression analysis on each of the outcome variables to account for potential confounding factors. This should be explained in the discussion.

I will answer to both concerns at once.

From a mathematical point of view, a MANOVA and a multivariate regression are basically the same (see Franco, 2019). Both belong to the family of general linear models. The main difference is that the independent variables used in the MANOVA must be categorial, whereas independent variables of regression models are allowed to be metric. When performing an experiment with an adequate number of participants (typically > 30 per condition), a general assumption is that potential moderators (i.e., confounding or influencing variables) will be equally distributed across the experimental conditions. This allows experimentally working researchers to draw causal inference that variations in the outcome variables are mainly due the experimentally varied factor (in this case scarcity versus surplus information). The statistical model of choice in this case is usually (M)ANOVA because the treatment variable is categorical. A regression, on the other hand, is particularly suitable when the influence of a variable is tested in interaction or with concurrent effects of many other variables. However, since no further control variables were collected in this study and the design is not correlative, I consider MANOVA to be indicated.

Currently, your methods section is a lengthy walk-through of the online survey. Some information you included, such as how participants were “thanked for their participation”, is not essential. Rather than describing the survey procedure in detail, please provide the well-articulated summary of what you were trying to measure, how they were measured in the survey (i.e. type of question or scale used), and how they were coded in the data. Also, whenever a technical term is introduced, spell it out the first time it was used, and explain what it is used for.

Also in order to meet previous points raised by the other reviewers, the method section was clarified and rephrased (

Page 5, sentence starting with “Invitation mails were sent out…”: What was the sampling pool for this invitation? what is meant by “weighted by” age and gender?

I revised this part in order to be more precise (l. 99-111):

“Participants were recruited between May 07 and 19, 2021 via a German crowdsourcing platform (Clickworker, 2021) and were compensated with EUR 0.60 for a maximum duration of 3 minutes. Invitation mails were sent out by the crowdsourcing platform and have been automatically weighted by the age and gender distribution of the German population. At the beginning of the online survey, participants confirmed the informed consent. As the treatment contained mild deception, the informed consent contained information that they might be deceived during participation and that the true purpose of the study could not be explained until the end.“

Page 6, paragraph starting with “Participants then answered questions…”: What are the dependent variables and what do you mean by ‘manipulation check’? What do you mean when participants were not directed to the “participation code”? What is the “debriefing information”? Are these important facts to be mentioned in this section?

I agree that this might have been not clear enough, so that I revised the section as follows (l. 129-135):

“Participants then answered questions on the dependent variables and the manipulation check, namely how good or bad they perceived the supply situation of their county. At the end of the survey, respondents were thanked for their participation and were informed of the intention of the study. Since deception took place, participants were not directed to the last survey page including the participation code for the payment until they indicated that they had read and understood the debriefing information. This has been implemented for ethical reasons, so that participants would be informed about the true purpose of the study in any case.“

Page 7, “Dependent variables”:

1) What was your rationale to include these three questions to measure the outcomes? Are these questions from a validated instrument to measure willingness to receive vaccination? (If yes, please cite the work.)

These items were developed in order to measure vaccination willingness in a specific study context. For this purpose, the items were not adapted from previous research, but self-developed. Vaccination willingness is often measured by a single item. However, there is rising debate of the use of single item measures in psychological research (see Allen et al., 2022 for a discussion). As this context was seen as one where multiple items could have been easily assessed, I therefore decided to use more than one item.

2) Why is the scale from 1 – 7? It appears to be unconventional.

Measuring vaccination willingness and other related concepts on a 7-point Likert scale is absolutely common (see Böhm et al., 2019; Huang et al., 2021; Sprengholz et al., 2021). 

3) Did you combine the three questions to form one outcome measure scale? If so, how did you combine them? (i.e., average of 3, sum of 3, etc.).

4) What is the ‘alpha’? (if it is Cronbach’s alpha, you need to specify it explicitly in the text and explain what it means)

Both points have been addressed by re-writing the section as follows (l. 127-135):

“Participants’ vaccination willingness was measured the mean value of the following three items: “Will you get vaccinated as soon as possible?”; “I will be trying harder to get a vaccination appointment in the near future.”; “If necessary, I will use various channels (e.g., primary physician and vaccination centers) to get a vaccination appointment.”; reaching from 1 = “very unlikely” to 7 = “very likely”, Cronbach’s α = .93, M = 4.65, SD = 1.98. The naming of the dependent variables differs from the preregistration. In the preregistration, vaccination willingness and appointment intention were mentioned separately. For reasons of conceptual overlap and high internal consistency, the items were combined into one scale.”

Page 7, “Manipulation check”:

What is the purpose of measuring this variable?

A manipulation check is a variable that measures whether the experimental treatment was successful or not, i.e., by measuring the mindset, feelings, cognitions, attitudes or what else was aimed to be manipulated by the experimental treatment (see Fiedler et al., 2022 for a concise discussion on the usefulness of manipulation checks in psychological research). 

Page 7, “Sample size and data analysis”

1) In the first sentence, please clarify what a “large effect size” is and a “small effect size” is. Please provide appropriate ranges.

To clarify the range of expected effect sizes, this part was revised as follows (l. 147-151):

“An a-priori power analysis using G*Power (version 3.1, Faul et al., 2009) for the detection of a medium (d = .50 to .70) to large (d = .80 to \\geq 1, Lenhardt & Lenhardt, 2016) scarcity effect in a Multivariate Analysis of Variance (MANOVA) including two dependent variables with d = .60, 1-β = .95, α = .05 resulted in a required sample size of 158 participants.”

2) For the power calculation, please specify d, 1-beta, alpha. Also, please spell-out MANOVA when it appears for the first time in the text.

This was already specified in the previous version, but I have also revised the part regarding the power analysis as follows (l. 144-151): 

“Since scarcity effects were found to be of large rather than small effect sizes in previous studies (see e.g., Jang et al., 2015; Verhallen, 1982; Worchel et al., 1975), a small sample size was assumed to be sufficient to identify scarcity effects on vaccination willingness. An a-priori power analysis using G*Power (version 3.1, Faul et al., 2009) for the detection of a medium (d = .50 to .70) to large (d = .80 to \\geq 1, Lenhardt & Lenhardt, 2016) scarcity effect in a Multivariate Analysis of Variance (MANOVA) including two dependent variables with d = .60, 1-β = .95, α = .05 resulted in a required sample size of 158 participants. To allow for potential dropout, data from 238 individuals was collected. A one-way MANOVA with treatment (scarcity vs. surplus) as an independent factor, and vaccination willingness and anger towards the debate over relaxations for vaccinated citizens as dependent variables was preregistered.”

3) In the last sentence, what do you mean when you say “A one-way MANOVA …. was pre-registered?”

As outlined in the Methods section of the manuscript, the study has been previously preregistered on aspredicted, in order to meet the criteria of open and transparent science (l. 91-93):

“Research question, design, minimum sample size, exclusion criteria, and analysis plan of the study were preregistered on AsPredicted (see https://aspredicted.org/blind.php?x=kt8us2).”

Page 7, “Exclusion criteria”

1) What is a relative speed index? Please describe.

This comment was addressed by re-writing the phrase (l. 161-164): 

“Following the recommendations of Leiner (2019a), participants with a high relative speed index (RSI > 2), which is a measurement provided by the survey platform used (sosci survey, Leiner, 2019b) and is based on processing time that identifies suspicious data patterns associated with poor quality, and reading times of equal to or less than 2 SD below the sample mean were excluded from data analysis.”

2) Spell out SD when it appears for the first time in the text.

SD means Standard Deviation and is part of the standard language when reporting scientific findings. According to the APA guidelines such everyday technical terms are not used to be spelled out.

3) The description of the final sample for the study should be provided in the “Results” section using a table including all the variables used in the analysis. Please provide such a table in this section.

In psychological research, the sample characteristics are typically reported in the Methods section. See Harris (2008) for a guide how to write up experiments in psychology or just the APA guidelines for reporting psychological research (see summary by Bhandari, 2022).

Results

1) Currently the results section lacks a descriptive table of the sample (a conventional table).

Again, this is not part of the Results section. However, an additional test was added before reporting the main analysis, in order to show that the samples of both treatment conditions were balanced with regard to the distribution of socio-demographic variables (l. 186-192):

“Overall, n = 98 respondents were assigned to the scarcity condition and n = 77 to the surplus condition. The sub-samples did not significantly differ regarding participants’ age, Mage scarcity = 36.02, SD = 9.80, Mage surplus = 35.81, SD = 10.05, t(173) = 0.14, p = .887, nor gender, χ2 (1, N = 175) = 0.23, p = .642, nor educational attainment, χ2 (6, N = 175) = 3.18, p = .786. With these results, systematic biases between treatment conditions can be excluded, which allows potential treatment effects to be attributed to the experimental manipulation.”

2) In addition, the way in which the results is reported in the text is hard to follow. A table for the results of the one-way MANOVA analysis for each group should be included. Author listed a list of abbreviated terms with figures without explaining what they mean. There is no effect size reported for the treatment, or no justification why the different tests were used to report the results.

According to the APA guidelines, tables should be avoided if they are unnecessary. As this is a short report, which reports effects on only two dependent variables, an in-text reference to the statistical findings seems to be more adequate. In addition, I always reported the effect sizes in the previous version of the manuscript (e.g., “Mscarcity = 4.91, SD = 1.88, Msurplus = 4.31, SD = 2.07, F(1,174) = 4.03, p = .043, η2 = .023“). η2 is a common effect size when reporting the results of a MANOVA.

Discussion

Overall, your discussion is based on the findings from your study to draw general inference on the German population. I am concerned on the validity of the discussion in general because 1) the generalizability of the findings based on a small sample of 175 participants is not discussed; and 2) the current study design does not allow any causal inference. Limitations should also expand further to accommodate the concerns raised in the “Methods” section.

I agree with the general concerns raised reviewer 6 regarding the interpretation and discussion of the effect found. I therefore established a more cautious tone and elaborated the weaknesses of the study and the need for further research in this area.

e.g., by changing the following parts in the discussion:

“This has left the impression that the scarcity of SARS-CoV-2 vaccines may increase the willingness of individuals to be vaccinated. However, this observation at the macro level did not allow for any causal conclusions on the micro level of vaccination behavior (Ramström, 2018). For this reason, an experimental study was conducted to test whether a scarcity impression of the SARS-CoV-2 vaccine can indeed lead to an increased willingness to vaccinate.“

“The results of the presented study support the idea that perceived scarcity of SARS-CoV-2 vaccines may be one driving factor for increasing people’s vaccination willingness. Of course, only limited conclusions about the complexity of vaccination decisions can be drawn from such a small study, and a cross-sectional experiment like this one only represents a momentary snapshot, through which it is not possible to construct a comprehensive explanation of the processes of the last weeks and months.”

Page 8, sentence starting with “The results support the hypothesis…”: Based on the current study design, there is very little evidence generated from the study to support this statement.

Based on the fact that the design and the main hypothesis of the paper were preregistered, I would not agree with the statement that there is little evidence based on the design. Especially through the experimental method underlying this study enables a strong test of the scarcity hypothesis.

Page 9, sentence starting with “Surprisingly, this has not..”: You cannot say this without testing for this. Your current study does not explore the association between the mentioned factors and your dependent variable.

I agree, this sentence was removed.

Page 9, sentence starting with “In fact, when some cities…”: This is irrelevant to the findings of this study and should rather presented in the introduction.

Page 9, sentence starting with “On the other hand, scarcity mentalities..”: This sentence is vague and does not add any value to the discussion as it is formed. Author should clarify it.

Page 9, sentence starting with ”Such anger may also quickly…”: Is there any literature documenting such an association? If yes, Author should cite it. This could be an interesting mechanism and Author should clearly state this in the introduction to justify the exploratory question (focusing on measurement of anger).

Thanks for these comments! I have revised the first part and second part criticized by the reviewer. Also I added a source for the claim I made on reactance.

„Such anger may also quickly turn into reactance and a reduced willingness to follow COVID-19 rules (Hajek & Häfner, 2021). This is also in line with findings of a later study by Sprangholz and colleagues (2022) showing that reactance towards COVID-19 measures was higher among people with a high willingness to vaccinate when they were told that vaccines would be scarce in the future.“

Minor revisions:

No lines numbers provided in the submission file. This made the review of this manuscript difficult which will also make its revision similarly difficult.

Referencing style does not comply with PLOS ONE’s submission guideline, and in-text citation format is inconsistent throughout (i.e., avoid adding hyperlink to a webpage directly in the text, use a citation software so that in-text citations are numbered in Vancouver style).

Footnotes are not permitted as per PLOS One’s guidelines.

Thanks for these helpful advices! I submitted the manuscript indicating that the correct style will be adapted at a later stage of the manuscript, as is done now.

I added lines to the manuscript and changed the citation style.

Overall, writing needs to be improved in terms of grammar and style. For example, inconsistent use of tense throughout the text; inconsistent use of terminology on “COVID-19 vaccine” (vs. SARS-CoV-2 vaccine)

Again, thanks for the advice. I have now consistently used the term “COVID-19” when referring to the COVID-19 pandemic and “SARS-CoV-2” to refer to the SARS-CoV-2 vaccine, as this is an adequate notation used by other authors in the field (e.g., Graeber et al., 2021). Also, the manuscript has gone through a native speaker’s (linguist’s) proof reading.

References

Allen, M. S., Iliescu, D., & Greiff, S. (2022). Single Item Measures in Psychological Science. European Journal of Psychological Assessment, 38, 1-5. https://doi.org/10.1027/1015-5759/a000699

Bhandari, Pritha (February 17, 2022). How to write an APA methods section with examples. https://www.scribbr.com/apa-style/methods-section/

Blanca Mena, M. J., Alarcón Postigo, R., Arnau Gras, J., Bono Cabré, R., & Bendayan, R. (2017). Non-normal data: Is ANOVA still a valid option?. Psicothema. 29(4). 552-557. Https://doi.org/10.7334/psicothema2016.383

Böhm, R., Meier, N. W., Groß, M., Korn, L., & Betsch, C. (2019). The willingness to vaccinate increases when vaccination protects others who have low responsibility for not being vaccinated. Journal of Behavioral Medicine, 42(3), 381-391. https://doi.org/10.1007/s10865-018-9985-9

Cortina, J. M. (1993). What is coefficient alpha? An examination of theory and applications. Journal of Applied Psychology, 78(1), 98–104. https://doi.org/10.1037/0021-9010.78.1.98

Draisma, S. & Dijkstra, W. (2004). Response latency and (para)linguistic expressions as indicators of response error. In S. Presser, M. P. Couper, J. T. Lessler, E. Mar- tin, J. Martin, J. M. Rothgeb, & E. Singer (Eds.), Wiley Series in Survey Methodology. Methods for testing and evaluating survey questionnaires. Hoboken, NJ: John Wiley & Sons.

Franco, G. (March 22, 2019). What is the difference between ANOVA and regression (and which one to choose). URL: https://www.statsimprove.com/en/what-is-the-difference-between-anova-and-regression-and-which-one-to-choose/

Graeber, D., Schmidt-Petri, C., & Schröder, C. (2021). Attitudes on voluntary and mandatory vaccination against COVID-19: Evidence from Germany. PloS one, 16(5), e0248372.

Huang, P. C., Hung, C. H., Kuo, Y. J., Chen, Y. P., Ahorsu, D. K., Yen, C. F., ... & Pakpour, A. H. (2021). Expanding protection motivation theory to explain willingness of COVID-19 vaccination uptake among Taiwanese university students. Vaccines, 9(9), 1046. https://doi.org/10.3390/vaccines9091046

Kidd, R. F. (1976). Manipulation checks: Advantage or disadvantage? Representative Research in Social Psychology, 7(2), 160–165.

Fiedler, K., McCaughey, L., & Prager, J. (2021). Quo vadis, methodology? The key role of manipulation checks for validity control and quality of science. Perspectives on Psychological Science, 16(4), 816-826. https://doi.org/10.1177/1745691620970602

Myers, A., & Hansen, C. H. (2012). Experimental psychology (7th edition). Belmont, CA: Wadsworth.

Nyhus Hagum, C., Tønnessen, E., & AI Shalfawi, S. (2022). Progression in training volume and perceived psychological and physiological training distress in Norwegian student athletes: A cross-sectional study. PloS one, 17(2), e0263575.

Roznik, E. A., & Alford, R. A. (2015). Seasonal ecology and behavior of an endangered rainforest frog (Litoria rheocola) threatened by disease. PLoS One, 10(5), e0127851.

Sprengholz, P., Betsch, C., & Böhm, R. (2021). Reactance revisited: Consequences of mandatory and scarce vaccination in the case of COVID‐19. Applied Psychology: Health and Well‐Being, 13(4), 986-995. https://doi.org/10.1111/aphw.12285

Tricarico, E., Borrelli, L., Gherardi, F., & Fiorito, G. (2011). I know my neighbour: individual recognition in Octopus vulgaris. PloS one, 6(4), e18710.

Warren, G. W., & Lofstedt, R. (2021). COVID-19 vaccine rollout risk communication strategies in Europe: a rapid response. Journal of Risk Research, 24(3-4), 369-379. https://doi.org/10.1080/13669877.2020.1870533

---

## [Decision Letter · Decision Letter 1]

24 Mar 2022

PONE-D-21-30612R1Does perceived scarcity of COVID-19 vaccines increase vaccination willingness? Results of an experimental study with German respondents in times of a national vaccine shortage.PLOS ONE

Dear Dr. Schnepf,

Thank you for submitting your manuscript to PLOS ONE. After careful consideration, we feel that it has merit but does not fully meet PLOS ONE’s publication criteria as it currently stands. Therefore, we invite you to submit a revised version of the manuscript that addresses the points raised during the review process.

ACADEMIC EDITOR: Still, reviewers are raising substantial concerns (reviewer # 6 is against publication) over the revised form of the MS. Do go through the comments and amend the MS accordingly. 

We look forward to receiving your revised manuscript.

Kind regards,

A. M. Abd El-Aty

Academic Editor

PLOS ONE

Reviewers' comments:

Reviewer's Responses to Questions

**Comments to the Author**

1. If the authors have adequately addressed your comments raised in a previous round of review and you feel that this manuscript is now acceptable for publication, you may indicate that here to bypass the “Comments to the Author” section, enter your conflict of interest statement in the “Confidential to Editor” section, and submit your "Accept" recommendation.

Reviewer #1: All comments have been addressed

Reviewer #2: All comments have been addressed

Reviewer #3: All comments have been addressed

Reviewer #4: (No Response)

Reviewer #5: All comments have been addressed

Reviewer #6: (No Response)

2. Is the manuscript technically sound, and do the data support the conclusions?

Reviewer #1: (No Response)

Reviewer #2: Yes

Reviewer #3: Yes

Reviewer #4: Partly

Reviewer #5: Yes

Reviewer #6: No

3. Has the statistical analysis been performed appropriately and rigorously? 

Reviewer #1: (No Response)

Reviewer #2: Yes

Reviewer #3: Yes

Reviewer #4: No

Reviewer #5: Yes

Reviewer #6: No

4. Have the authors made all data underlying the findings in their manuscript fully available?

Reviewer #1: (No Response)

Reviewer #2: Yes

Reviewer #3: Yes

Reviewer #4: Yes

Reviewer #5: Yes

Reviewer #6: No

5. Is the manuscript presented in an intelligible fashion and written in standard English?

Reviewer #1: (No Response)

Reviewer #2: Yes

Reviewer #3: Yes

Reviewer #4: Yes

Reviewer #5: Yes

Reviewer #6: No

6. Review Comments to the Author

Reviewer #1: (No Response)

Reviewer #2: - the author should be add the list of abbreviation and types of statistical software used for analysis

Reviewer #3: (No Response)

Reviewer #4: The author does a reasonable job in addressing many of the issues I raised, especially regarding providing additional information about the experiment. If other reviewers and the editors are ok with a paper that has little to say about mechanisms and external validity, I do not want to stand in the way of its publication. However, there are two issues with the empirical analysis that I have major concerns about, and cannot support the publication of a paper on a topic with important public health implications in a wide-read scientific journal until they are addressed. At the moment, the results are not credible enough in my view. Fortunately, there are simple fixes that I propose below.

First, I remain seriously concerned that the exclusion of people that the manipulation did not work for is introducing bias by conditioning the sample on a function of treatment. Specifically, the article notes that: “participants for whom the scarcity vs. surplus manipulation did not work, i.e., individuals in the scarcity condition who perceived the availability to be high (perceived vaccine availability > 4) and individuals in the surplus condition who perceived the availability to be low (perceived vaccine availability < 4), were excluded from the data analysis.” Although the preregistration stated “participants for whom the treatment failed”, it did not prespecify how this would be done.

This is simply bad practice, and seriously risks generating biased estimates. Let me explain. If the baseline mean is below 4 (which it seems to be), there is a risk that the experimental design is broken because participants in the scarcity treatment is less likely to be dropped because they are more likely to have a score below 4 than participants surplus treatment are likely to have a score above 4. Indeed, this fits with the fairly substantial imbalance across scarcity (N=98) and surplus conditions (N=77) after this exclusion criteria is applied, despite even probabilities of treatment assignment (as I understand it). I imagine that this 44% to 56% split would be statistically distinguishable from 50:50 in even this small sample. The consequence of this is that the randomization is likely to be undone because people in the treatment and control groups possess different baseline beliefs due to the exclusion criterion; this may in turn drive differences in observed outcomes. The paper now conducts balance tests over age, gender, and education, and finds no difference; while this is somewhat comforting, these variables are not necessarily correlated with the outcomes and inspection of the tables in the supplementary materials do suggest quite large differences in education in magnitude.

Together, these concerns create significant worries about the validity of the results – especially for journal like PLoS One and on an important public health issue.

Fortunately, there is a simple fix here: not imposing this exclusion criteria (although the others are fine). This will immediately restore the experimental properties of the analysis. If the author is concerned about observing small effects because some participants did not internalize the treatment, they should conduct an instrumental variables analysis in addition to reporting the reduced form comparison between treatment and control. Given the simplicity of the solution and obvious risk of bias, I cannot support publication of this article until this more sensible analysis is used for the main results. This is not simply a robustness check, but the proper experimental specification should be the main result.

Second, and less important, the key dependent variables deviate from the preregistered approach in two respects: (i) there is no mention of creating a mean scale; (ii) one outcome (pursuing various channels) included in the scale in the paper is not mentioned at all in the preregistered analysis. (This is not the exploratory outcome.) This obviously creates concerns regarding researcher degrees of freedom.

To be more transparent in the presentation of the results, I would like the author to report the effect of manipulation on each of the three outcomes separately (i.e. not just as part of the scale, although it’s ok to keep that). This accords with the method they preregistered. This is important because the outcome that is not mentioned in the preregistration could be driving the results, and seems less conceptually relevant for the study. This is also a simple fix.

Ultimately, I strongly encourage the author to implement these two simple changes that would substantially enhance the credibility of the study’s results. Absent these changes, I would continue to have serious concerns about the validity of the paper’s conclusions.

Reviewer #5: I have no further comments. All of my suggestions have been addressed by the author. The document includes now all of the details about the statistical analysis that were not clear before.

Reviewer #6: The author partially, but not sufficiently, addressed the raised points. The manuscript reads much better now with clear description of the methodology and better presentation of the results. I understand the author’s choice of MANOVA given that sociodemographic characteristics were not significantly different across the two groups, therefore were assumed to be relatively well randomized. However, some of the points raised originally on the manuscript still remain unaddressed. At this point, I would not see the current manuscript fit for publication in PLOS ONE for its audience. I would recommend authors to seek for publication opportunities in psychology-centric journals.

Author consistently rebutted the concerns on the description of the methodology referring to psychology literatures/textbooks. However, given the main audience of PLOS ONE is interdisciplinary and the topic is targeting at public health professionals, I still see the choice of “experimental study” and “treatment” inappropriate. Specifically, the author did not respond to the raised concern on whether the instant exposure to the crafted information on vaccine surplus/scarcity can be considered an appropriate exposure to the treatment given that the immediate (and prolonged) change in participants’ perception about vaccination given this exposure is unlikely.

Also, the fact that the study protocol, or its simple summary, was registered to the website the author indicated does not provide any justification on the validity of the hypothesis or the strength of the results. In fact, if this is a true experimental study involving a treatment/intervention on the human subjects, this should have been registered to a proper trial registry (https://www.who.int/clinical-trials-registry-platform/network/trial-registration)

The author also claimed the superiority of the experimental study design, however, the author missed the important reason why it is considered superior to the cross-sectional correlational data (i.e. temporality). The pre-study comparison of sociodemographic characteristics is helpful, but is not sufficient to establish a difference between the two groups in the perception towards COVID-19 vaccine is established post treatment / exposure. To ensure the quantification of a causal impact, ensuring the temporality, pre-post comparison is recommended on the same outcome variable.

I still strongly recommend the author to include a table on descriptive statistics. For any observational study (following the STROBE guideline) or randomized controlled trial study (following CONSORT guidelines), the inclusion of descriptive statistics is strongly recommend/required. Author kept insisting it is a “short report”, however there is no submission category for short reports of such kind. If this is to be considered under the research article category, minimum reporting requirements should be met.

7. PLOS authors have the option to publish the peer review history of their article (what does this mean?). If published, this will include your full peer review and any attached files.

Reviewer #1: **Yes: **Sarah J.E. Barry

Reviewer #2: No

Reviewer #3: No

Reviewer #4: No

Reviewer #5: No

Reviewer #6: No

---

## [Author Response · Author response to Decision Letter 1]

9 May 2022

Reviewer #4: The author does a reasonable job in addressing many of the issues I raised, especially regarding providing additional information about the experiment. If other reviewers and the editors are ok with a paper that has little to say about mechanisms and external validity, I do not want to stand in the way of its publication. However, there are two issues with the empirical analysis that I have major concerns about, and cannot support the publication of a paper on a topic with important public health implications in a wide-read scientific journal until they are addressed. At the moment, the results are not credible enough in my view. Fortunately, there are simple fixes that I propose below.

First, I remain seriously concerned that the exclusion of people that the manipulation did not work for is introducing bias by conditioning the sample on a function of treatment. Specifically, the article notes that: “participants for whom the scarcity vs. surplus manipulation did not work, i.e., individuals in the scarcity condition who perceived the availability to be high (perceived vaccine availability > 4) and individuals in the surplus condition who perceived the availability to be low (perceived vaccine availability < 4), were excluded from the data analysis.” Although the preregistration stated “participants for whom the treatment failed”, it did not prespecify how this would be done.

I want to answer to this concern by drawing a broader picture on the nature of experimental research in psychology/behavioral economics/communication science. The basic idea of an experiment is that a potential causal relationship between two variables x (explanans) and y (explanandum) is tested by varying different levels of x, and randomly assigning them to the participants. Afterwards, it is tested whether these different levels of x resulted in different degrees of y. In the case of the underlying study x is scarcity versus surplus information and y is vaccination willingness and anger about relaxations. Different from medical research, for instance, in psychological/behavioral economics/communication science research, experimental conditions typically aim to (temporally) change or affect participants’ perceptions of a situation, their cognitive states, as well as feelings, thoughts or mindsets about something. However, the success of the experimental manipulation, which often is done by just exposing participants to a short newspaper expert, is conditioned to so many external variables, such as whether participants draw attention to the presented piece of information, whether they have fully read the information (processing time), whether they have understood it correctly, and – very importantly – whether they believe in the information or not. As the presented research dealt with a highly topical and widely discussed issue, and thereby aimed to increase the ecological validity and applicability of the study, a test of whether the experimental manipulation worked among participants or not was more than necessary. In the preregistration form this was indicated by defining the exclusion criterion of excluding those for whom the manipulation did not work. I think from a pure logical perspective, this means the experimental manipulation worked for participants in the scarcity condition who perceived the availability of COVID-19 vaccines to be bad and participants in the surplus condition who perceived the availability to be good. In contrast, the manipulation failed for participants in the scarcity condition who perceived the availability of COVID-19 vaccines to be good and participants in the surplus condition who perceived the availability to be bad. Empirically, as a logical consequence, if the manipulation check variable of perceived vaccine availability reaches from 1 (very bad) to 7 (very good), a failed manipulation is operationalized by:

Participants in the scarcity condition who perceive the availability to be good, i.e., higher than 4 (scale mean).

Participants in the surplus condition who perceive the availability to be bad, i.e., lower than 4 (scale mean).

This is simply bad practice, and seriously risks generating biased estimates. Let me explain. If the baseline mean is below 4 (which it seems to be), there is a risk that the experimental design is broken because participants in the scarcity treatment is less likely to be dropped because they are more likely to have a score below 4 than participants surplus treatment are likely to have a score above 4. Indeed, this fits with the fairly substantial imbalance across scarcity (N=98) and surplus conditions (N=77) after this exclusion criteria is applied, despite even probabilities of treatment assignment (as I understand it). I imagine that this 44% to 56% split would be statistically distinguishable from 50:50 in even this small sample. The consequence of this is that the randomization is likely to be undone because people in the treatment and control groups possess different baseline beliefs due to the exclusion criterion; this may in turn drive differences in observed outcomes. 

There is a lot of high impact literature claiming that manipulation checks in experimental research are more than ever needed, especially behind the background of the replication crisis in psychological and behavioral science (e.g., Fiedler et al., 2021; Gollwitzer & Schwabe, 2021). The experimental design is not „broken" by this practice, but is guaranteed by the manipulation check in the first place. Making use of the manipulation check as an exclusion criterion (which was preregistered) does not confound the results, but in the opposite, it saves the study from being confounded by other background variables, such as high knowledge about the actual supply situation or string trust in the responsible local authorities to deal with the situation. Only by implementing the manipulation check as an exclusion criterion it is guaranteed that observed changes in the dependent variable can be attributed to previous variations of the independent variable. Or to say it with the words of Fiedler et al. (2021, p. 818):

“MCs are critical for the viability of the logical premise of a theoretical hypothesis H: Δx → Δy, which predicts a shift in the dependent variable (DV) y (Δy) given a shift in independent variable (IV) x (Δx). This prediction is logically contingent on the premise that an experimental treatment actually succeeds in inducing the intended Δx shift. Without that premise, predicting an effect Δy is unwarranted.” 

In a similar vein, Gollwitzer and Schwabe (2021) argue that manipulation checks are particularly recommended for the use of subtle manipulations (p. 6).

I don’t share R4’s assessment in this aspect. However, I outlined in more detail and with a stronger reference to existing research, why I used the manipulation check as an exclusion criterion and how this can help to increase data quality (p. 7).

“There is a growing body of criticism on doing experimental research without controlling for a successful experimental manipulation within the sample under research [32-33]. Given that people’s prior knowledge and information could have strongly influenced whether people believed the treatment texts or not, we preregistered the manipulation check variable as an exclusion criterion, as done in other experimental research [34-36].”

Fiedler, K., McCaughey, L., & Prager, J. (2021). Quo vadis, methodology? The key role of manipulation checks for validity control and quality of science. Perspectives on Psychological Science, 16(4), 816-826. https://doi.org/10.1177/1745691620970602

Gollwitzer, M., & Schwabe, J. (2020). Context dependency as a Predictor of Replicability. Review of General Psychology, 10892680211015635. https://doi.org/10.1177/10892680211015635

The paper now conducts balance tests over age, gender, and education, and finds no difference; while this is somewhat comforting, these variables are not necessarily correlated with the outcomes and inspection of the tables in the supplementary materials do suggest quite large differences in education in magnitude.

In contrast to the above-mentioned criticism on the use of a manipulation check as an exclusion variable, testing the sample for being balanced in sociodemographic variables was very useful and a good recommendation, as especially age and gender are strong predictors of individuals’ vaccination willingness (Coscia, 2021). With regard to the distribution of the sociodemographics across the experimental conditions, I also wrote in the manuscript (p. 8) that the cell differences in educational attainment between experimental conditions were not statistically significant.

“The sub-samples did not significantly differ regarding participants’ age, Mage scarcity = 36.02, SD = 9.80, Mage surplus = 35.81, SD = 10.05, t(173) = 0.14, p = .887, nor gender, χ2 (1, N = 175) = 0.23, p = .642, nor educational attainment, χ2 (6, N = 175) = 3.18, p = .786.”

Nevertheless, regarding the overall bias in gender, i.e., having more male participants in the sample than females, which is in contrast to the gender distribution of the unvaccinated population at the field time of the study, during which according to Coscia (2021) more women were hesitant to vaccinate against COVID-19 than men, I reanalyzed the data by using a weighting variable to correct the oversampling of male non-vaccinated compared to females. According to the data reported by Coscia for Germany, I used the following weighting procedure:

As, according to Coscia (2021), 53.11 % of vaccine sceptics were female and 46.89 % were male in the broad time range of the study. The actual gender distribution in the sample was 38.3 % female and 61.7 % male. 

The weighting variable was calculated following the classical weighting process recommended by IBM SPSS by dividing the target distribution through the actual distribution, resulting in:

Weighting factor for female participants: 53.11 / 38.3 = 1.39

Weighting factor for male participants: 46.89 / 61.7 = 0.76

The results remained the same when using this weighting variable, which was now also mentioned in the manuscript (p. 9).

“(…) and when the sample was weighted for participants gender, using a weighting variable which represents the actual gender distribution among vaccination sceptics at the field time of the survey (supplement S4).”

Coscia, Verena (October 6, 2021). Vaccination willingness in Europe: Who are the unvaccinated? Max Planck Institute for Social Law and Social Policy. https://www.mpisoc.mpg.de/en/newsroom/news/detail/announce/vaccination-willingness-in-europe-who-are-the-unvaccinated/

Together, these concerns create significant worries about the validity of the results – especially for journal like PLoS One and on an important public health issue.

In PLoS One more than 45,600 hits appear when searching for articles reporting manipulation checks in their materials and methods section. More than 8,000 hits appear in PloS One when “manipulation check” is paired with “exclusion criteria”. Many of these articles reported that the manipulation check (using different paradigms) was used as an exclusion criterion, e.g.,

“An additional 32 participants who failed the memory manipulation check (more than 1 incorrect choice)”

Sanchez, C., Sundermeier, B., Gray, K., & Calin-Jageman, R. J. (2017). Direct replication of Gervais & Norenzayan (2012): No evidence that analytic thinking decreases religious belief. PloS One, 12(2), e0172636.

“Nineteen patients failed the manipulation check (that is responded to the item “I was ignored” with 1 or 2), and were therefore excluded from the following analyses. The final sample hence comprised 97 patients.”

De Rubeis, J., Sütterlin, S., Lange, D., Pawelzik, M., van Randenborgh, A., Victor, D., & Vögele, C. (2016). Attachment status affects heart rate responses to experimental ostracism in inpatients with depression. PloS One, 11(3), e0150375.

“Accordingly, complete data from 43 participants were available for analysis (although only 41 were included in the final analysis, due to 2 participants failing the manipulation check—see Self Report Measures).”

Almarzouki, A. F., Brown, C. A., Brown, R. J., Leung, M. H., & Jones, A. K. (2017). Negative expectations interfere with the analgesic effect of safety cues on pain perception by priming the cortical representation of pain in the midcingulate cortex. Plos One, 12(6), e0180006.

“Not all participants from the sampling procedure were included in the experiment. Since the study was conducted online using a self-administered survey experiment with video messages, we administered the manipulations carefully to ensure exposure. One hundred twenty-five respondents, homogenously distributed across conditions, were excluded for one of three reasons: not being able to hear the audio check, reporting having been exposed to the video in full outside of the study, or not being able to play the experiment video: (1) 85 reported they could not hear, and others entered the wrong number during an audio check at the beginning of the study. The 85 participants who could not hear were asked if they had another audio-working device. Of those 85, the 64 who said they did not were dropped from the experiment; others joined with a new device. (2) After viewing the video, 32 respondents stated that they had seen it fully somewhere and were thus dropped from analysis because prior exposure would confound our experimental manipulation. However, 80 respondents who said they remembered seeing part of the video were retained. (3) 22 participants indicated they could not view the entire video and were dropped from analysis due to inadequate exposure to the experimental manipulation. (4) 7 participants who indicated they could not hear the experiment video were dropped as well due to inadequate exposure to the experimental manipulation. Additionally, those who left the audio check question blank (N = 56) at the very beginning of the survey were excluded.

Due to a survey progression (display logic) error, some participants were able to answer questions without having been shown the experimental module; this reduction in sample size was also random across conditions. Hence, the overall experiment sample size was 2,345. A small number of participants with missing data in dependent and independent variables were also dropped. Final Ns are reported along with the results in Table 1.”

Kuru, O., Stecula, D., Lu, H., Ophir, Y., Chan, M. P. S., Winneg, K., ... & Albarracín, D. (2021). The effects of scientific messages and narratives about vaccination. PloS One, 16(3), e0248328.

I think these examples speak for themselves and rather show that manipulation checks are often used to improve data quality and are by no means used otherwise. 

Fortunately, there is a simple fix here: not imposing this exclusion criteria (although the others are fine). This will immediately restore the experimental properties of the analysis. If the author is concerned about observing small effects because some participants did not internalize the treatment, they should conduct an instrumental variables analysis in addition to reporting the reduced form comparison between treatment and control. Given the simplicity of the solution and obvious risk of bias, I cannot support publication of this article until this more sensible analysis is used for the main results. This is not simply a robustness check, but the proper experimental specification should be the main result.

I completely see it the other way around. As outlined above, analyzing the data without guaranteeing that the experimental manipulation has worked for the subjects under study would be – as demonstrated above – not reasonable. It is more than clear that the effect diminishes when participants are included who were exposed to scarcity information but did not believe it neither for participants who were in the surplus condition and did not believe in this information. This is simply a cutoff of research with high ecological validity which is situated in a current context. There one must simply exclude participants who do not believe in the presented information. Otherwise, the sub-samples are highly confounded. This is not a disadvantage of the conducted research, but a circumstance to which applied research is inevitably exposed.

Second, and less important, the key dependent variables deviate from the preregistered approach in two respects: (i) there is no mention of creating a mean scale; (ii) one outcome (pursuing various channels) included in the scale in the paper is not mentioned at all in the preregistered analysis. (This is not the exploratory outcome.) This obviously creates concerns regarding researcher degrees of freedom.

I am sorry, the preregistration file just misses an “e.g.,” when I referred to vaccination willingness as it is one line below when I mentioned appointment intention. I can understand the reviewer’s criticism, but I think I have mentioned this satisfyingly in the last revision of the manuscript. As there was a high conceptual overlap between appointment intentions and general vaccination willingness, I already outlined why I found it useful to summarize the items. In addition, there is some criticism to use single item measures when multi-item measures are available (e.g., Fisher et al., 2016). I also outlined this in the current revision (see p. 6).

“In addition, the items were summarized into one scale as the use of single item measures is increasingly criticized if multi-item measures are possible [28].”

Even though I can understand the criticism, I want to say that I cannot do more than being transparent with the deviations from the preregistration – which I am in the whole manuscript.

Fisher, G. G., Matthews, R. A., & Gibbons, A. M. (2016). Developing and investigating the use of single-item measures in organizational research. Journal of Occupational Health Psychology, 21(1), 3–23. https://doi.org/10.1037/a0039139

To be more transparent in the presentation of the results, I would like the author to report the effect of manipulation on each of the three outcomes separately (i.e. not just as part of the scale, although it’s ok to keep that). This accords with the method they preregistered. This is important because the outcome that is not mentioned in the preregistration could be driving the results, and seems less conceptually relevant for the study. This is also a simple fix.

I already did this at a former stage of the review process (see page 9 of the last revised version)!

Ultimately, I strongly encourage the author to implement these two simple changes that would substantially enhance the credibility of the study’s results. Absent these changes, I would continue to have serious concerns about the validity of the paper’s conclusions.

I tried to outline above why I disagree with some of the reviewer’s recommendations.

Reviewer #6: The author partially, but not sufficiently, addressed the raised points. The manuscript reads much better now with clear description of the methodology and better presentation of the results. I understand the author’s choice of MANOVA given that sociodemographic characteristics were not significantly different across the two groups, therefore were assumed to be relatively well randomized. However, some of the points raised originally on the manuscript still remain unaddressed. At this point, I would not see the current manuscript fit for publication in PLOS ONE for its audience. I would recommend authors to seek for publication opportunities in psychology-centric journals.

Author consistently rebutted the concerns on the description of the methodology referring to psychology literatures/textbooks. However, given the main audience of PLOS ONE is interdisciplinary and the topic is targeting at public health professionals, I still see the choice of “experimental study” and “treatment” inappropriate. 

Yes, I referred to textbooks and basic literature in the last round of reviews several times, because I obviously had to explain basic knowledge about the rationale of experiments, which was very unfortunate for a peer-review process in an international journal.

I'm irritated by R6's comment that this research is not supposed to fit into the PloS One scope. As R6 him/herself mentions, PloS One is an interdisciplinary journal that has a high profile ESPECIALLY in Psychology and Health Communication. For the keywords vaccination willingness alone, more than 25,000 hits can be found at PloS One! Some of the articles from PloS One cited in my manuscript are dedicated to a very similar topic. Therefore, I see R6's comment as unwarranted and unsupported.

Specifically, the author did not respond to the raised concern on whether the instant exposure to the crafted information on vaccine surplus/scarcity can be considered an appropriate exposure to the treatment given that the immediate (and prolonged) change in participants’ perception about vaccination given this exposure is unlikely.

I don’t see this comment well established, as I have discussed this point authoritatively in the revised version of the manuscript and in the previous response letter, justifying why the experimental manipulation can be considered warranted and that other research has now used similar manipulations of vaccine scarcity – even in PloS One.

Also, the fact that the study protocol, or its simple summary, was registered to the website the author indicated does not provide any justification on the validity of the hypothesis or the strength of the results. In fact, if this is a true experimental study involving a treatment/intervention on the human subjects, this should have been registered to a proper trial registry (https://www.who.int/clinical-trials-registry-platform/network/trial-registration)

I clearly have to disagree with the reviewer. More than 71,000 hits appear in PloS One when I search for articles including the keywords “registered the open science framework”! The Open Science Framework is the best-known Open Science Platform in Cognitive and Behavioral Sciences and is a critical part of the Open Science movement (Open Science Collaboration, 2015). This study is not a clinical trial study!!! I checked the link and this recommendation is completely inappropriate as it is related to medical/clinical research – this manuscript refers to opinion measures not clinical measures.

Open Science Collaboration. (2015). Estimating the reproducibility of psychological science. Science, 349(6251), aac4716.

The author also claimed the superiority of the experimental study design, however, the author missed the important reason why it is considered superior to the cross-sectional correlational data (i.e. temporality). The pre-study comparison of sociodemographic characteristics is helpful, but is not sufficient to establish a difference between the two groups in the perception towards COVID-19 vaccine is established post treatment / exposure. To ensure the quantification of a causal impact, ensuring the temporality, pre-post comparison is recommended on the same outcome variable.

I already explained this in the last round of reviews. Cross-sectional / correlational data must NEVER be interpreted causally! In contrast, experimental research allows for an adequate test of whether an independent variably causally affects a dependent variable. Also, a pre-post design is a repetition of measurement but per definitionem not necessarily an experiment! I have the impression that the reviewer mixes some things up, here.

I still strongly recommend the author to include a table on descriptive statistics. For any observational study (following the STROBE guideline) or randomized controlled trial study (following CONSORT guidelines), the inclusion of descriptive statistics is strongly recommend/required. Author kept insisting it is a “short report”, however there is no submission category for short reports of such kind. If this is to be considered under the research article category, minimum reporting requirements should be met.

A table of the descriptive statistic was already part of the revised manuscript (see p. 7 and Supplement 1 of the last revision)!

---

## [Decision Letter · Decision Letter 2]

18 May 2022

PONE-D-21-30612R2Does perceived scarcity of COVID-19 vaccines increase vaccination willingness? Results of an experimental study with German respondents in times of a national vaccine shortage.PLOS ONE

Dear Dr. Schnepf,

Thank you for submitting your manuscript to PLOS ONE. After careful consideration, we feel that it has merit but does not fully meet PLOS ONE’s publication criteria as it currently stands. Therefore, we invite you to submit a revised version of the manuscript that addresses the points raised during the review process.

ACADEMIC EDITOR: Still the reviewers are raising substantial concern on the revised form of the MS. Would you please go through the comments and amend the MS accordingly.==============================

We look forward to receiving your revised manuscript.

Kind regards,

A. M. Abd El-Aty

Academic Editor

PLOS ONE

Reviewers' comments:

Reviewer's Responses to Questions

**Comments to the Author**

1. If the authors have adequately addressed your comments raised in a previous round of review and you feel that this manuscript is now acceptable for publication, you may indicate that here to bypass the “Comments to the Author” section, enter your conflict of interest statement in the “Confidential to Editor” section, and submit your "Accept" recommendation.

Reviewer #2: All comments have been addressed

Reviewer #4: (No Response)

2. Is the manuscript technically sound, and do the data support the conclusions?

Reviewer #2: Yes

Reviewer #4: Partly

3. Has the statistical analysis been performed appropriately and rigorously? 

Reviewer #2: Yes

Reviewer #4: No

4. Have the authors made all data underlying the findings in their manuscript fully available?

Reviewer #2: Yes

Reviewer #4: Yes

5. Is the manuscript presented in an intelligible fashion and written in standard English?

Reviewer #2: Yes

Reviewer #4: Yes

6. Review Comments to the Author

Reviewer #2: - the author should include lists of abbreviations

-the author should lists source of data, method of data collection, statistical model and statistical software used for analysis

Reviewer #4: Thank you for the detailed response to my comments, and those of reviewer 6. I think the author has made useful progress in this revision. However, the important point about introducing bias is still not properly addressed. I expand on why below. I also think that the author can still be clearer about what is and is not registered in the main paper, but this is a more minor concern.

ISSUE 1: INTRODUCING BIAS BY POST-TREATMENT SAMPLE SELECTION

I remain unsatisfied with the author’s response on my most important point regarding the individuals that were dropped from the sample. While I appreciate the author’s extended response and I appreciate that there are differences across disciplines, everything in the modern causal inference toolkit in econometrics and program evaluation says that restricting the sample on the basis of post-treatment outcome is not good practice and can introduce bias. Indeed, Aronow, Baron, and Pinson (2019) prove exactly this in the context of dropping subjects based on manipulation tests in a recent article in Political Analysis. Quoting from the introduction of their article: “…we show that the practice of dropping subjects based on a manipulation check should generally be avoided. We provide a number of statistical results establishing that doing so can bias estimates or undermine identification of causal effects. We also show that this practice is equivalent to inducing differential attrition across treatment arms, which may induce bias of unknown sign and magnitude. We do not claim that our statistical formulations are particularly novel—they follow from well-known results about conditioning on post-treatment variables and attrition—but, given the prevalence of this practice, we believe that the relationship between these findings and practice in experimentation is underappreciated.” In the context of this paper, my previous review provided an example of how such a bias could arise in this particular application. It also noted that there is a notable difference in the number of treated and control units that emerges after the sample is trimmed, which suggests the risk of bias.

The author did not seriously engage with this point and seemed to miss my point about manipulation tests. Everyone, including myself, agrees that manipulation tests are useful for validating that the treatment did what it intended to do. There is no disagreement here – with either the author or their quote from Fiedler et al. (2021) or their reference to Gollwitzer and Schwabe (2021) in their response. However, as just explained, dropping observations can bias experimental comparisons when treated and control units are dropped in differential ways, which may occur when they are dropped on the basis of post-treatment outcomes like manipulation checks. I am not saying that there is necessarily bias in this particular case, but there is reason to believe there could be – as explained above and in my previous review.

The author goes on to state that many other papers also drop observations based on the results of manipulation tests. (Note though that many attention tests are not post-treatment or plausibly unrelated to treatment, which makes bias unlikely.) However, I do not believe that others also following bad practice is sufficient reason to risk biasing experimental designs by dropping data on the basis of post-treatment variables – especially when the results could influence policy decisions, and particularly when better methods exist and could easily be implemented instead. Following the seminal econometric textbook by Angrist and Pischke (2008) that builds on Angrist’s Nobel prize-winning work on instrumental variables, I continue to recommend the following:

1. First conduct a reduced form analysis in the *entire* sample that compares treated and control units. The outcomes would be for the manipulation test outcome and the primary outcomes used in the paper. This would yield the average treatment effect. To be clear, this estimate is not confounded; sure, people might not react to treatment for many reasons, but a successful randomization ensures that this is balanced across treatment groups.

2. If treatment compliance is low, as the author notes, then the natural next step is to use an *instrumental variables* analysis in the full sample. This retains the underlying randomization as the driver of differences in outcomes across treatment groups, but scales up the coefficients to adjust for non-compliance. Under an exclusion restriction (treatment does not affect the outcome except through posterior beliefs) and imposing monotonicity (that treatment conditions cause people to update in a given direction), this also has a causal interpretation – the local average treatment effect among compliers. As far as I understand, this is the author’s target estimand; the sample restriction is designed to estimate effects only for the compliers.

If the results are similar, great – this suggests that the author’s method is not introducing significant bias; then I think it’s fine to report these approaches as robustness checks. If the results differ substantially, there is serious reason to worry about the internal validity of the reported results. In that case, I would favor estimates from the specifications I just proposed, because they do not risk post-treatment bias.

It's also worth noting that the reweighting approach is useful, and helps to make the sample a bit more representative. However, it doesn’t directly address the concern raised - potential imbalances across treatment conditions with respect to observable pre-treatment covariates. Still, as the author notes, there are not significant differences here, even if some differences look somewhat large in magnitude (but perhaps lack of significance might reflect the small sample).

ISSUE 2: TRANSPARENCY ABOUT PREREGISTERED OUTCOMES

With respect to deviations from the pre-analysis plan, I apologize for missing the results in appendix S3. These results do, however, show that the largest effects are on the outcome that was not preregistered, while effects are borderline or null for the two outcomes that were preregistered. Regardless of whether the preregistration plan simply forgot to include this outcome (or an “e.g.”), I think the author should clearly state in the main paper that the index results are driven by an outcome that was not preregistered. I have no problem with publishing finding for outcomes that are not preregistered, but this should be clearly communicated. Readers can make up their mind about whether to downweight the results, but in this case I agree with the author that this outcome is relevant and should be included in the index.

7. PLOS authors have the option to publish the peer review history of their article (what does this mean?). If published, this will include your full peer review and any attached files.

Reviewer #2: **Yes: **Yenew Alemu Mihret

Reviewer #4: No

---

## [Author Response · Author response to Decision Letter 2]

5 Jul 2022

Dear Dr. A. M. Abd El-Aty, 

Thank you for your invitation of revising and resubmitting the manuscript titled “Does perceived scarcity of COVID-19 vaccines increase vaccination willingness? Results of an experimental study with German respondents in times of a national vaccine shortage.”

I have revised the last version of the manuscript in accordance to many of the reviewers’ comments.

Regarding some concerns on the methodological appropriateness, I have different opinions as one of the reviewers, on which I want to reply in more detail in this response letter. More precisely, I have undertaken several analyses to make sure that the chosen exclusion criteria have not led to any bias between included and excluded participants. I have reported these additional analyses in the supplements of the manuscripts, following the CONSORT flow-chart.

In the following, I want to outline point by point, how I addressed the reviewers’ concerns. Therefore, I printed my answers in bold:

Reviewer #2: - the author should include lists of abbreviations

-the author should lists source of data, method of data collection, statistical model and statistical software used for analysis

A list of abbreviations is now attached in the notes section of the manuscript.

Reviewer #4: Thank you for the detailed response to my comments, and those of reviewer 6. I think the author has made useful progress in this revision. However, the important point about introducing bias is still not properly addressed. I expand on why below. I also think that the author can still be clearer about what is and is not registered in the main paper, but this is a more minor concern.

ISSUE 1: INTRODUCING BIAS BY POST-TREATMENT SAMPLE SELECTION

I remain unsatisfied with the author’s response on my most important point regarding the individuals that were dropped from the sample. While I appreciate the author’s extended response and I appreciate that there are differences across disciplines, everything in the modern causal inference toolkit in econometrics and program evaluation says that restricting the sample on the basis of post-treatment outcome is not good practice and can introduce bias. 

Sorry, but I have to disagree. This is not the one and only current opinion in econometrics and program evaluation which the reviewer is describing here. As I outlined in my last response, this method is quite often practiced (also in PLOS One) and after my (and other authors’) opinion, the aptness of post-treatment selection widely depends on questions like how is the treatment induced and how can its success be proven/measured? I don’t agree that there is an absolute “the-one-and-only-approach” to handle exclusion criteria. As outlined below, I am able to show that there are no significant differences between the sub-samples regarding their socio-demographic composition, so I would claim that Aronow’s et al. (2019) argumentation of possible biases does not apply to the analyses undertaken in this manuscript. 

Indeed, Aronow, Baron, and Pinson (2019) prove exactly this in the context of dropping subjects based on manipulation tests in a recent article in Political Analysis. Quoting from the introduction of their article: “…we show that the practice of dropping subjects based on a manipulation check should generally be avoided. We provide a number of statistical results establishing that doing so can bias estimates or undermine identification of causal effects. We also show that this practice is equivalent to inducing differential attrition across treatment arms, which may induce bias of unknown sign and magnitude. We do not claim that our statistical formulations are particularly novel—they follow from well-known results about conditioning on post-treatment variables and attrition—but, given the prevalence of this practice, we believe that the relationship between these findings and practice in experimentation is underappreciated.” In the context of this paper, my previous review provided an example of how such a bias could arise in this particular application. It also noted that there is a notable difference in the number of treated and control units that emerges after the sample is trimmed, which suggests the risk of bias.

The author did not seriously engage with this point and seemed to miss my point about manipulation tests. Everyone, including myself, agrees that manipulation tests are useful for validating that the treatment did what it intended to do. There is no disagreement here – with either the author or their quote from Fiedler et al. (2021) or their reference to Gollwitzer and Schwabe (2021) in their response. However, as just explained, dropping observations can bias experimental comparisons when treated and control units are dropped in differential ways, which may occur when they are dropped on the basis of post-treatment outcomes like manipulation checks. I am not saying that there is necessarily bias in this particular case, but there is reason to believe there could be – as explained above and in my previous review.

The author goes on to state that many other papers also drop observations based on the results of manipulation tests. (Note though that many attention tests are not post-treatment or plausibly unrelated to treatment, which makes bias unlikely.) However, I do not believe that others also following bad practice is sufficient reason to risk biasing experimental designs by dropping data on the basis of post-treatment variables – especially when the results could influence policy decisions, and particularly when better methods exist and could easily be implemented instead. Following the seminal econometric textbook by Angrist and Pischke (2008) that builds on Angrist’s Nobel prize-winning work on instrumental variables, I continue to recommend the following:

1. First conduct a reduced form analysis in the *entire* sample that compares treated and control units. The outcomes would be for the manipulation test outcome and the primary outcomes used in the paper. This would yield the average treatment effect. To be clear, this estimate is not confounded; sure, people might not react to treatment for many reasons, but a successful randomization ensures that this is balanced across treatment groups.

[As I will outline in more detail below, there is no observable confound or bias between the included and excluded individuals.]

2. If treatment compliance is low, as the author notes, then the natural next step is to use an *instrumental variables* analysis in the full sample. This retains the underlying randomization as the driver of differences in outcomes across treatment groups, but scales up the coefficients to adjust for non-compliance. Under an exclusion restriction (treatment does not affect the outcome except through posterior beliefs) and imposing monotonicity (that treatment conditions cause people to update in a given direction), this also has a causal interpretation – the local average treatment effect among compliers. As far as I understand, this is the author’s target estimand; the sample restriction is designed to estimate effects only for the compliers.

If the results are similar, great – this suggests that the author’s method is not introducing significant bias; then I think it’s fine to report these approaches as robustness checks. If the results differ substantially, there is serious reason to worry about the internal validity of the reported results. In that case, I would favor estimates from the specifications I just proposed, because they do not risk post-treatment bias.

It's also worth noting that the reweighting approach is useful, and helps to make the sample a bit more representative. However, it doesn’t directly address the concern raised - potential imbalances across treatment conditions with respect to observable pre-treatment covariates. Still, as the author notes, there are not significant differences here, even if some differences look somewhat large in magnitude (but perhaps lack of significance might reflect the small sample).

I would like to note here that Aronow et al. (2019) argue from a very different perspective. Namely, that it cannot be ruled out that a treatment also influences those participants who failed a manipulation check. In my opinion, this is highly dependent on the variables a researcher examines. For example, if an experimental manipulation aims at mood change, it may be that a manipulation check would exclude a person who is below a certain threshold that the researchers consider to be the minimum necessary mood change. However, the person may have experienced a significant change for him or her intra-individually despite a small change in mood. Their exclusion from the overall sample would thus lead to an underestimation of the global effect and indicate a bias in the sample. Aronow et al. (2019) draw attention to exactly this problem and show, doing a replication of a study without using exclusion criteria, that the effect becomes stronger. In addition, they argue that using manipulation checks as an exclusion criterion can lead to a serious problems and biases when comparing experimental treatment groups including a MC with control conditions not including a MC. 

In my opinion, however, such generalizations (“Manipulation checks are always inappropriate as inclusion/exclusion criteria.”) do not apply to all types of experimental manipulations. Whether a MC is useful as an exclusion variable depends on the treatment, the measurement of the MC, and the dependent variables. If it is important for the validity of a treatment that the participants process a given text information correctly or believe the information that is presented, then it makes no sense to include persons in the analysis who do not believe the information. If the MC guarantees that participants have read, understood, and believe the content of a treatment text, for example, then that is a legitimate inclusion variable. Why should the treatment have any effect on people who do not believe the treatment text? 

Of course, this tendency is also evident in my data. If people for whom the MC failed are included in the analyses, then there is no longer a significant treatment effect. But that is perfectly reasonably, because then there are many participants in the sample who may not have read the text, understood it, or simply do not believe the given information. Thus, it makes no sense to report the conduct the analyses over the total sample. 

In addition, another point made by Aronow et al. (2019) and by Reviewer 4 is that the use of MCs as exclusion criteria can lead to biases between the sub-samples. I checked this statistically and I can say based on statistical comparisons between included and excluded participants that they are not significantly different from each other in socio-demographic variables. In addition, there was no difference between the sub-samples, when checking for interactions between the exclusion criterion and the experimental condition. I also included this note after reporting the CONSORT chart and subsample characteristics in the Supplement:

[To test whether there is potential bias through the chosen exclusion criteria, it was tested whether the excluded and included sub-samples significantly differed in their sociodemographic composition. An ANOVA with two factors (experimental condition: scarcity versus surplus; filter: included versus excluded) for testing on age differences between the sub-samples revealed no significant difference between the included and excluded participants, Mincluded = 35.93, SD = 9.88, Mexcluded = 35.22, SD = 11.33, F(1, 138) = 0.26, p = 607, η^2 = .001. In addition, the interaction between the filter variable and the experimental condition was non-significant, so no age bias was detected between the sub-samples, F(1, 138) = 0.09, p = 759, η^2 < .001. For gender and educational attainment, two Chi-square tests were conducted to test for group differences. Again, there was no difference between the included and excluded participants with regard to gender, χ^2(N = 238) = 0.06, p = .803, and educational attainment, χ^2(N = 238) = 3.16, p = .788. Also, there was no significant interaction effect between the filter variable and the experimental condition. So, there was no difference between the sub-samples with regard to gender, 〖χ^2〗_scarcity(N = 119) = 0.50, p = .477, 〖χ^2〗_surplus(N = 119) < 0.01, p = .982, and educational attainment, 〖χ^2〗_scarcity(N = 119) = 1.15, p = .949, 〖χ^2〗_surplus(N = 238) = 2.43, p = .907.]

Since this is a criticism where the opinions of reviewer 4 and myself differ widely, I can do nothing else but point out that no relevant bias can be found in my sub-samples and that I have tried to take up reviewer 4’s criticism by focusing more on the weaknesses of the treatment in the discussion and on the fact that more field studies are needed to be able to prove such effects more robustly.

[See p. 11-12, l. 253-273: “The results of the presented study support the idea that perceived scarcity of SARS-CoV-2 vaccines may be one driving factor for increasing people’s vaccination willingness. Given the simplicity of the design and the rather small size of the effects found, of course, only limited conclusions about the complexity of vaccination decisions can be drawn from such a small study, and a cross-sectional experiment like this one only represents a momentary snapshot, through which it is not possible to construct a comprehensive explanation of the processes of the last weeks and months. Against this background, it also has to be mentioned here that a significant number of participants was excluded as the experimental manipulation has not worked for them, i.e., their perceived availability of COVID-19 vaccines was not affected by the experimental condition. On one hand, using manipulation checks as an exclusion criterion guarantees that the participants under research have been influenced by the given information in the intended direction, which is a necessary precondition to draw any causal conclusions between the experimental variation and the variation on the dependent variables [32-33]. On the other hand, this practice is increasingly criticized in some social science disciplines [43] because of the risk of increasing biases between the sub-samples. As in this study there were no significant differences between the tested sub-samples regarding their sociodemographic composition, the risk of potential bias in this study is comparatively small. Nevertheless, it has to be critically mentioned here that the high number of excluded cases due to a failed manipulation check may speak for the fact that some participants might not have completely read or understood the information, or that they were highly informed about the issue so that the presented information probably has not affected their perception of vaccine availability. This is a clear limitation of the presented study and can only be solved in the future by (a) choosing more distinctive texts to induce scarcity or surplus impressions, (b) conducting quasi-experimental field experiments in which local scarcity situations are covered as independent variables, as well as (c) cross-sectional research on the effects of temporal vaccine scarcity on individuals’ vaccination willingness.”]

ISSUE 2: TRANSPARENCY ABOUT PREREGISTERED OUTCOMES

With respect to deviations from the pre-analysis plan, I apologize for missing the results in appendix S3. These results do, however, show that the largest effects are on the outcome that was not preregistered, while effects are borderline or null for the two outcomes that were preregistered. Regardless of whether the preregistration plan simply forgot to include this outcome (or an “e.g.”), I think the author should clearly state in the main paper that the index results are driven by an outcome that was not preregistered. I have no problem with publishing finding for outcomes that are not preregistered, but this should be clearly communicated. Readers can make up their mind about whether to downweight the results, but in this case I agree with the author that this outcome is relevant and should be included in the index.

I addressed these points and made this clear in the manuscript.

[p. 7, l. 135-141: “The naming of the dependent variables differs from the preregistration. In the preregistration, vaccination willingness and appointment intention were mentioned separately. For reasons of conceptual overlap and high internal consistency, the items were combined into one general vaccination willingness scale, as the use of single item measures is increasingly criticized if multi-item measures are possible [28]. Also, item three was not explicitly mentioned in the preregistration as the preregistration only included a reference to example items of each scale.”]

---

## [Decision Letter · Decision Letter 3]

25 Jul 2022

PONE-D-21-30612R3Does perceived scarcity of COVID-19 vaccines increase vaccination willingness? Results of an experimental study with German respondents in times of a national vaccine shortage.PLOS ONE

Dear Dr. Schnepf,

Thank you for submitting your manuscript to PLOS ONE. After careful consideration, we feel that it has merit but does not fully meet PLOS ONE’s publication criteria as it currently stands. Therefore, we invite you to submit a revised version of the manuscript that addresses the points raised during the review process.

ACADEMIC EDITOR: The reviewer remain concerned that the findings are not robust. Would you please go through the comments and provide the requested data/results by the diligent reviewer to consolidate the findings.==============================

We look forward to receiving your revised manuscript.

Kind regards,

A. M. Abd El-Aty

Academic Editor

PLOS ONE

Reviewers' comments:

Reviewer's Responses to Questions

**Comments to the Author**

1. If the authors have adequately addressed your comments raised in a previous round of review and you feel that this manuscript is now acceptable for publication, you may indicate that here to bypass the “Comments to the Author” section, enter your conflict of interest statement in the “Confidential to Editor” section, and submit your "Accept" recommendation.

Reviewer #2: All comments have been addressed

Reviewer #4: (No Response)

2. Is the manuscript technically sound, and do the data support the conclusions?

Reviewer #2: Yes

Reviewer #4: Partly

3. Has the statistical analysis been performed appropriately and rigorously? 

Reviewer #2: Yes

Reviewer #4: No

4. Have the authors made all data underlying the findings in their manuscript fully available?

Reviewer #2: Yes

Reviewer #4: Yes

5. Is the manuscript presented in an intelligible fashion and written in standard English?

Reviewer #2: Yes

Reviewer #4: Yes

6. Review Comments to the Author

Reviewer #2: I have seen the previous manuscript before the authors sent rivised manuscript.all comments are edited. This manuscript should be publish. It is appropriate for publication.

Reviewer #4: I appreciate the author’s detailed response, and I regard the additional text re the second issue (the outcome that was not preregistered) as resolved. However, I remain concerned that the estimation strategy could be generating biased results.

The author is right that including inattentive respondents would drag down the ATE relative to the ATE for a fully attentive sample, assuming that inattentive respondents experience similar treatment effects as attentive respondents. The author further notes: “If people for whom the MC failed are included in the analyses, then there is no longer a significant treatment effect.” Given this, we cannot tell whether the results go away in the full sample because it includes people that don’t respond to treatment or because the exclusion criteria based on post-treatment manipulation tests introduces bias because, as Aronow et al. (2019) show, this practice can break the randomization.

The natural way to address the author’s point that “there are many participants in the sample who may not have read the text, understood it, or simply do not believe the given information” is to conduct an *instrumental variables* analysis. This will address this issue by estimating the ATE among manipulation check *compliers*. Intuitively, this will rescale the reduced form coefficient to produce what I believe to be the estimand the author care about, but without inducing the risk of bias by differentially dropping treated and control respondents. I have suggested IV approach in all three of my reviews, but the author still has not addressed this suggested solution, which is standard practice for addressing non-compliance.

(Beyond a randomized experiment, an IV analysis imposes two further identifying assumptions: monotonicity and the exclusion restriction. The former is surely satisfied, unless the treatment caused people to believe the opposite of the prompt they received - which would be a bigger problem, but seems unlikely, as the manipulation check suggests. The exclusion restriction requires that the treatment does not affect the outcome except when the manipulation check works. This is a stronger assumption, but would be satisfied if - as the author claims - individuals only did not pass the manipulation because they did not sufficiently engage with the treatment to be affected by it.)

In the absence of an IV analysis, which addresses the problem of potentially biased exclusion criteria while adjusting for non-compliance (without dropping observations), it is hard to be confident in the author’s findings. I recognize that there are no imbalances after imposing the exclusion criteria on the three covariates (age, gender, and education) that the author measured. (While comparing the excluded and included samples is an informative diagnostic, the most relevant test here compares the included sample across treatment conditions – i.e. the balance tests that have always been in the manuscript.) However, there could still be imbalances on many other unobservables. There is reason to worry about this because notably more participants drop from the surplus conditions (final sample of 77) than the scarcity condition (final sample of 98). If there was no such differential attrition (instead resulting in an approximate 50:50 split), I would be not be as concerned about the risk of undoing the random assignment.

If the author is not going to conduct the instrumental variables analysis, I am happy for the editors to adjudicate on this important statistical issue that seems to affect the core findings.

Following these comments, I believe two minor elements of the manuscript remain inaccurate. First is the CONSORT diagram: since the manipulation check occurred *after* treatment assignment and depended on engagement with treatment, a separate box for these exclusions should occur in the branch for each treatment condition. This would clarify for reader the issue that the author and I have debated extensively. Second, the claim that “Overall, n = 98 respondents were assigned to the scarcity condition and n = 195 77 to the surplus condition.” on p9 is misleading. These are not the totals assigned to treatment, but the totals after applying the exclusion criteria.

7. PLOS authors have the option to publish the peer review history of their article (what does this mean?). If published, this will include your full peer review and any attached files.

Reviewer #2: **Yes: **Yenew Alemu

Reviewer #4: No

---

## [Author Response · Author response to Decision Letter 3]

28 Jul 2022

Reviewer #2: I have seen the previous manuscript before the authors sent rivised manuscript.all comments are edited. This manuscript should be publish. It is appropriate for publication.

I thank Reviewer 2 very much for his/her recommendation of publication!

Reviewer #4: I appreciate the author’s detailed response, and I regard the additional text re the second issue (the outcome that was not preregistered) as resolved. However, I remain concerned that the estimation strategy could be generating biased results.

The author is right that including inattentive respondents would drag down the ATE relative to the ATE for a fully attentive sample, assuming that inattentive respondents experience similar treatment effects as attentive respondents. The author further notes: “If people for whom the MC failed are included in the analyses, then there is no longer a significant treatment effect.” Given this, we cannot tell whether the results go away in the full sample because it includes people that don’t respond to treatment or because the exclusion criteria based on post-treatment manipulation tests introduces bias because, as Aronow et al. (2019) show, this practice can break the randomization.

As I have previously shown in several comparisons between the sub-samples (excluded versus included etc.), there is no evidence for a break of the randomization in this specific study.

The natural way to address the author’s point that “there are many participants in the sample who may not have read the text, understood it, or simply do not believe the given information” is to conduct an *instrumental variables* analysis. This will address this issue by estimating the ATE among manipulation check *compliers*. Intuitively, this will rescale the reduced form coefficient to produce what I believe to be the estimand the author care about, but without inducing the risk of bias by differentially dropping treated and control respondents. I have suggested IV approach in all three of my reviews, but the author still has not addressed this suggested solution, which is standard practice for addressing non-compliance.

I don’t see the beneficial additional information of estimating the average treatment effect (ATE) among participants for whom the MC worked or not. In addition, having read some articles on the estimation of the ATE, I don’t really see how this should help to disentangle the potential reasons of failed manipulation checks in my data or how to increase the information on the effects found in study. However, I performed this analysis.

Typically, the ATE estimation is used to compare participants who experienced any form of a treatment (e.g., participants in an experimental condition) or who have a certain characteristic and those who haven’t (e.g., participants in a control group). Mostly, this procedure is used if a researcher assumes an omitted variable to influence the results. For example, if participants with a university degree (Y1) and without a university degree (Y0) are compared regarding their income level, one omitted variable behind this process could be their general ability, i.e., their choice to do so or not. Thus, the ATE procedure would try to disentangle between criterion (university degree = 1 versus no university degree = 0) and choice (choosing to do a university degree = 1 or not = 0). As data on such omitted variables is mostly missing, ATE estimation is often based on naïve estimators for cells without real data. This is why I struggled to some extent to transfer the ATE estimation to our discussion on the role of the manipulation check. 

The ATE is defined as the difference between the outcome variable among individuals who received a treatment and those who haven’t. 1/N ∑_i▒〖〖(Y〗_1 (i)-Y_0 (i))〗. I transferred the idea of the ATE estimation on the underlying phenomenon of the manipulation check as follows. Y1(i) = person i’s outcome (vaccination willingness / anger) if he/she is in the scarcity condition; Y0(i) = person i’s outcome (vaccination willingness / anger) if he/she is in the surplus condition

Here I see the first problem: In this study, I don’t compare a treatment group with a control group, so I am not convinced that this procedure is fully apt.

Nevertheless, if this is defined as Y1 and Y0, then the ATE = E(Y1) – E(Y0). E = expected value.

Now, we can take the manipulation check into account by defining the ATT and ATN. The ATT is defined as the ATE among the “treated” individuals, in this case participants for whom the manipulation check worked (MC = 1). The ATN is the ATE among the “non-treated” individuals which I translated to participants for whom the manipulation check didn’t work (MC = 0). This results in the 2x2 matrices below.

Values of the outcome variable “vaccination willingness” among different subgroups

 MC = 0 MC = 1

Y0 5.33 4.31

Y1 5.22 4.91

Now, the ATT would be 4.91 – 4.31 = 0.60 and the ATN would result in 5.22 – 5.33 = -0.11.

A positive value indicates that scarcity information increased vaccination willingness, a negative value indicates a decrease. If these differences are tested against zero, this results in t(155.64) = 1.84, p = .076 [p < .10 counts as statistical evidence for one-sided tests] for the ATT and t(60) = 0.21, p = .836 for the ATN.

Values of the outcome variable “anger” among different subgroups

 MC = 0 MC = 1

Y0 4.56 4.05

Y1 4.52 5.25

Now, the ATT would be 4.75 – 4.05 = 0.70 and the ATN would result in 4.52– 5.56 = -0.04.

A positive value indicates that scarcity information increased anger, a negative value indicates a decrease. If these differences are tested against zero, this results in t(172) = 2.07, p = .040 for the ATT and t(60) = 0.37, p = .836 for the ATN.

(Beyond a randomized experiment, an IV analysis imposes two further identifying assumptions: monotonicity and the exclusion restriction. The former is surely satisfied, unless the treatment caused people to believe the opposite of the prompt they received - which would be a bigger problem, but seems unlikely, as the manipulation check suggests. The exclusion restriction requires that the treatment does not affect the outcome except when the manipulation check works. This is a stronger assumption, but would be satisfied if - as the author claims - individuals only did not pass the manipulation because they did not sufficiently engage with the treatment to be affected by it.)

I think that results outlined above clearly support the idea that the experimental manipulation has affected individuals with a successful MC but not those with a failed MC. So, it can be excluded, that individuals with a failed MC were affected (in any direction) by the treatment.

In the absence of an IV analysis, which addresses the problem of potentially biased exclusion criteria while adjusting for non-compliance (without dropping observations), it is hard to be confident in the author’s findings. I recognize that there are no imbalances after imposing the exclusion criteria on the three covariates (age, gender, and education) that the author measured. (While comparing the excluded and included samples is an informative diagnostic, the most relevant test here compares the included sample across treatment conditions – i.e. the balance tests that have always been in the manuscript.) However, there could still be imbalances on many other unobservables. 

I see Reviewer 4’s point here. However, the thing is these potential variables were (as he/she correctly states) unobserved. So, I don’t see how I should be able to test any additional differences such as regarding information seeking etc.

I have added a lot of possibilities and alternative explanations to the discussion section after the last rounds of revisions.

There is reason to worry about this because notably more participants drop from the surplus conditions (final sample of 77) than the scarcity condition (final sample of 98). If there was no such differential attrition (instead resulting in an approximate 50:50 split), I would be not be as concerned about the risk of undoing the random assignment.

I already adjusted for this imbalance by using weighting procedures, and the findings kept stable. When I weighted the cases for a 50:50 distribution, the scarcity effect becomes even more pronounced (see figure below).

Results of the MANOVA across the weighted sample 

In addition, regarding the socio-demographic composition, there is no indicator of a potential bias (which could be, for instance, indicated by differences in educational attainment – which is not the case, as I have added in one of the last revisions). Also, there is no hint that participants for whom the MC failed resulted reported a particularly low vaccination willingness or any “suspicious pattern” indicating that the exclusion criterion could have inherently affected the findings. As visible by the graphics below, vaccination willingness (anger) was even slightly higher in this group.

If the author is not going to conduct the instrumental variables analysis, I am happy for the editors to adjudicate on this important statistical issue that seems to affect the core findings.

I have now somehow translated the desired instrumental variables analyses in form of an additional mediation model between Treatment - Perceived vaccine availability (manipulation check variable) - Vaccination willingness / Anger (see Table 1 and 2). However, I do not feel that this additional analysis would improve the manuscript in any way. I rather have the impression that it is still not a good idea to run an analysis across the entire sample. 

After having read different articles on the instrumental variable approach, there are several assumptions underlying this analysis. 

1) the analysis is especially recommended when regression models are performed and a predictor is correlated with the error term. � I don’t see why this should be apt for my analysis. I have performed an experiment, so I experimentally varied the independent variable. Thus, I don’t see why Reviewer 4 assumes an endogeneity problem. 

2) The instrumental variable should be uncorrelated with the error term. � If Reviewer 4 thinks that problem 1 is given in my data, I don’t see which instrumental variable should solve this, as I don’t see that there is any instrumental variable that meets the criterion of being fully „exogenous“ in my data set. 

3) If the IV approach is used to deal with omitted variables, the author should have an idea of which omitted variable may have caused a finding. � If Reviewer 4 assumes omitted variables, I don’t see how I should be able to identify them within the data I have. 

Thus, given the underlying data structure and the experimental method, I don’t see that this approach is fully apt for this study. Given my data, I can just say, that the manipulation has worked for some people, but not for others. And that it does only make sense from a causal inference perspective to include those for whom the manipulation has worked.

Table 1

Results of the mediation model predicting vaccination willingness by treatment via perceived vaccine availability (manipulation check).

 Perceived vaccine availability

 B SE t p 95% CI

Constant 3.59 .09 40.39 <. 001 3.41, 3.76

Treatment 

(-1 = Surplus, +1 = Scarcity) -0.22 .09 -2.43 .016 -0.39, -0.04

 Vaccination willingness

Constant 5.32 .35 15.25 < .001 4.63, 6.01

Treatment 

(-1 = Surplus, +1 = Scarcity) 0.12 .12 0.97 .335 -0.12, 0.37

Perceived vaccine availability -0.14 0.09 -1.54 .125 -0.32, 0.04

Note. a-path: R2 = .02, F(2,234) = 5.91, p = .016; b-path: R2 = .02, F(2,233) = 1.93, p = .147

Total effect: c = 0.15, SE = .12, t = 1.22, p = .224, 95% CI [-0.09, 0.39]

Indirect effect: ab = 0.03, Boot SE = .02, Boot CI [-0.01, 0.01]

N = 238

Table 2

Results of the mediation model predicting anger by treatment via perceived vaccine availability (manipulation check).

 Perceived vaccine availability

 B SE t p 95% CI

Constant 3.59 .09 40.39 <. 001 3.41, 3.76

Treatment 

(-1 = Surplus, +1 = Scarcity) -0.22 .09 -2.43 .016 -0.39, -0.04

 Anger

Constant 5.33 .38 14.11 < .001 4.59, 6.08

Treatment 

(-1 = Surplus, +1 = Scarcity) 0.19 .13 1.40 .164 -0.08, 0.46

Perceived vaccine availability -0.24 0.10 -2.44 .015 -0.43, -0.05

Note. a-path: R2 = .02, F(2,234) = 5.91, p = .016; b-path: R2 = .02, F(2,233) = 1.93, p = .147

Total effect: c = 0.24, SE = .13, t = 1.78, p = .076, 95% CI [-0.02, 0.51]

Indirect effect: ab = 0.05, Boot SE = .03, Boot CI [0.01, 0.12]

N = 238

Following these comments, I believe two minor elements of the manuscript remain inaccurate. First is the CONSORT diagram: since the manipulation check occurred *after* treatment assignment and depended on engagement with treatment, a separate box for these exclusions should occur in the branch for each treatment condition. This would clarify for reader the issue that the author and I have debated extensively. Second, the claim that “Overall, n = 98 respondents were assigned to the scarcity condition and n = 77 to the surplus condition.” on p9 is misleading. These are not the totals assigned to treatment, but the totals after applying the exclusion criteria.

I really don’t see what I should change here. 

With regard to the CONSORT diagram, I can just emphasize that I used the CONSORT diagram in accordance with the recommendations on the CONSORT homepage, so I don’t see where I should change it.

I have added every information on every sub-sample (included, excluded, condition etc.). I have also added the information on the group comparison between both experimental groups consisting in the final sample in the main manuscript and the (insignificant) differences between excluded and included sub-samples (indicating no sub-sample bias) in the supplemental materials to which I reference in the main text. I feel that it would be misleading to report the whole sub-sample comparisons in more detail in the main manuscript, as the results focus on the final sample and not the entire sample.

---

## [Editor Report · Decision Letter 4]

9 Aug 2022

Does perceived scarcity of COVID-19 vaccines increase vaccination willingness? Results of an experimental study with German respondents in times of a national vaccine shortage.

PONE-D-21-30612R4

Dear Dr. Schnepf,

We’re pleased to inform you that your manuscript has been judged scientifically suitable for publication and will be formally accepted for publication once it meets all outstanding technical requirements.

Kind regards,

A. M. Abd El-Aty

Academic Editor

PLOS ONE

Additional Editor Comments (optional):

The authors respond satisfactorily to the comments raised by the reviewer
---

## [Editor Report · Acceptance letter]

11 Aug 2022

PONE-D-21-30612R4 

Does perceived scarcity of COVID-19 vaccines increase vaccination willingness? Results of an experimental study with German respondents in times of a national vaccine shortage. 

Dear Dr. Schnepf:

I'm pleased to inform you that your manuscript has been deemed suitable for publication in PLOS ONE. Congratulations! Your manuscript is now with our production department. 

Kind regards, 

on behalf of

Prof. A. M. Abd El-Aty 

Academic Editor

PLOS ONE